# Therapeutic Efficacy of Interferon-Gamma and Hypoxia-Primed Mesenchymal Stromal Cells and Their Extracellular Vesicles: Underlying Mechanisms and Potentials in Clinical Translation

**DOI:** 10.3390/biomedicines12061369

**Published:** 2024-06-20

**Authors:** Yu Ling Tan, Maimonah Eissa Al-Masawa, Sue Ping Eng, Mohamad Nasir Shafiee, Jia Xian Law, Min Hwei Ng

**Affiliations:** 1Centre for Tissue Engineering and Regenerative Medicine, Faculty of Medicine, Universiti Kebangsaan Malaysia, Jalan Yaacob Latif, Bandar Tun Razak, Kuala Lumpur 56000, Malaysia; tan970812@gmail.com (Y.L.T.); maimonah.almasawa@gmail.com (M.E.A.-M.); lawjx@ppukm.ukm.edu.my (J.X.L.); 2NK Biocell Sdn. Bhd, Unit 1-22A, 1st Floor Pusat Perdagangan Berpadu (United Point), No.10, Jalan Lang Emas, Kuala Lumpur 51200, Malaysia; sueping_86@yahoo.com; 3Department of Obstetrics & Gynaecology, Faculty of Medicine, Universiti Kebangsaan Malaysia, Kuala Lumpur 56000, Malaysia; nasirshafiee@ukm.edu.my

**Keywords:** mesenchymal stromal cells, extracellular vesicles, priming, interferon-gamma, hypoxia, pre-clinical

## Abstract

Multipotent mesenchymal stromal cells (MSCs) hold promises for cell therapy and tissue engineering due to their self-renewal and differentiation abilities, along with immunomodulatory properties and trophic factor secretion. Extracellular vesicles (EVs) from MSCs offer similar therapeutic effects. However, MSCs are heterogeneous and lead to variable outcomes. In vitro priming enhances MSC performance, improving immunomodulation, angiogenesis, proliferation, and tissue regeneration. Various stimuli, such as cytokines, growth factors, and oxygen tension, can prime MSCs. Two classical priming methods, interferon-gamma (IFN-γ) and hypoxia, enhance MSC immunomodulation, although standardized protocols are lacking. This review discusses priming protocols, highlighting the most commonly used concentrations and durations, along with mechanisms and in vivo therapeutics effects of primed MSCs and their EVs. The feasibility of up-scaling their production was also discussed. The review concluded that priming with IFN-γ or hypoxia (alone or in combination with other factors) boosted the immunomodulation capability of MSCs and their EVs, primarily via the JAK/STAT and PI3K/AKT and Leptin/JAK/STAT and TGF-β/Smad signalling pathways, respectively. Incorporating priming in MSC and EV production enables translation into cell-based or cell-free therapies for various disorders.

## 1. Introduction

Multipotent mesenchymal stromal cells (MSCs) are adult stem cells that can self-renew and differentiate into various mesenchymal cell lineages, including osteocytes, adipocytes, and chondrocytes [1]. The high self-renewal capacity in vitro, multi-lineage differentiation potential, trophic factor secretion, and immunomodulatory properties of MSCs have made them popular biological candidates in cell therapies and tissue engineering in the past 30 years [2]. Furthermore, the low expression of CD40, CD80, CD86, and major histocompatibility complex class I (MHC I), as well as the lack of MHC II expression, have rendered them immuno-privileged and therefore a highly valuable allogeneic cell therapy tool for regenerative medicine [3]. MSCs were initially discovered in the bone marrow, but they have now been discovered in other tissues, including adipose tissue, muscle, peripheral blood, hair follicles, teeth, placenta, umbilical cord, and umbilical cord blood [1,4,5,6]. MSCs derived from different tissue sources displayed varying properties in terms of cellular composition (varying cell phenotypes, i.e., surface markers and immune profile), lineage-specific differentiation potential, and self-renewal capacities [7]. In addition to their regenerative properties, MSCs exhibit robust systemic immunosuppressive effects via various mechanisms. Consequently, cell therapy based on MSCs is viewed as a promising approach for addressing autoimmune or inflammatory conditions, including graft-versus-host disease (GVHD), inflammatory bowel disease, multiple sclerosis, and coronavirus disease 2019 (COVID-19) [8].

Key mechanisms underlying MSC-based therapy primarily include three aspects. Firstly, cell replacement involves MSCs differentiating into various cell types to replace damaged tissues, integrating into affected areas. Secondly, immunomodulation occurs via MSCs’ paracrine and extracellular vesicle secretion, regulating immune responses. Lastly, cell rescue involves MSCs transferring their organelles to injured cells via diverse mechanisms such as direct cell-to-cell contact and cell fusion [9]. MSCs are known to exert their paracrine effects via a secretome comprising various soluble factors and extracellular vesicles (EVs), crucial for intercellular communication and signalling. Suppression of EV secretion resulted in MSCs losing their immunomodulatory effect [10]. Furthermore, EVs’ payload mirrors the molecular and functional properties of their producing cells [11]. Although MSC therapies show promising therapeutic results in preclinical studies and clinical trials for a number of diseases, there are some challenges faced in the clinical applications of MSCs that have yet to be addressed. The inconsistency of MSCs in terms of immune compatibility, stability, heterogeneity, differentiation, and migratory capability are just a few of the factors that have contributed to the failure of MSC clinical development [12]. The traits of MSCs are shaped by in vivo and in vitro biological, biochemical, and biophysical factors, which closely govern their functionality and survival. To date, numerous studies have shown that altering biological, biochemical, and biophysical factors can affect the fate of MSCs, their lineage-specific differentiation capability, and their functionality, eventually affecting their therapeutic potential in regenerative medicine [13,14,15].

According to Wang et al., MSCs are inert and incapable of dampening immune responses unless prompted by specific combinations of inflammatory cytokines [16]. Cell priming, also known as preconditioning or licensing [7], is one of the commonly used strategies to augment MSC functionality and their EVs. Generally, MSCs can be primed with cytokines, hypoxia, growth factors, pharmacological or chemical agents, biomaterials, and different culture conditions [17]. The characteristics of inflamed tissue, including low oxygen tension, elevated concentrations of inflammatory cytokines, and the presence of microorganisms, significantly influence the metabolism and functions of cells at the injury site. In these challenging environments, transplanted MSCs are expected to modulate the inflammatory and immune responses and facilitate the regenerative process. Consequently, there is a growing interest in preconditioning MSCs to enhance their physiology and fortify their therapeutic mechanisms. This approach aims to boost the number of cells reaching the site of inflammation, improve their survival, and enhance their anti-inflammatory and regenerative effects [18].

Priming MSCs with cytokines like interferon-gamma (IFN-γ), tumour necrosis factor-alpha (TNF-α), interleukin 1 alpha (IL-1α), and interleukin 1 beta (IL-1β) have been widely utilized to augment their immunomodulatory potential. This pre-activation enables MSCs to produce various functional factors that exert specific immunomodulatory effects [3]. Notably, IFN-γ and TNF-α have emerged as the predominant cytokines linked to MSC stimulation. However, in comparison, IFN-γ has been identified as a more potent inducer of immunoregulatory factors in MSCs, such as prostaglandin E_2_ (PGE_2_), indoleamine 2,3-dioxygenase (IDO), and hepatocyte growth factor (HGF), compared to TNF-α [19]. MSCs primed with IFN-γ lead to the upregulation of immunomodulatory molecules, including IDO, PGE_2_, and transforming growth factor-beta (TGF-β), as well as increased expression of class I and class II histocompatibility leucocyte antigen (HLA) molecules and co-stimulatory molecules [20]. Zhang et al. showed that even exosomes derived from IFN-γ-primed MSCs suppressed inflammatory cell infiltration, promoted angiogenesis, and reduced apoptosis in a rat model of myocardial injury [21]. Furthermore, Zhang et al. demonstrated that pre-treating MSCs with IFN-γ and TNF-α effectively eliminated donor-dependent variations in immunomodulation [22].

In the human body, oxygen availability in the tissues depends on the extent of vascularization and metabolic requirement of the tissue, often lower than the atmospheric oxygen levels (21%) [7]. In the physiological state, the oxygen level in peripheral tissues, referred to as ‘physioxia’, typically ranges between 1% and 11% [23], and they physiologically adapt to this condition [7]. In fact, MSC expansion in normoxic cell culture conditions (21%) may lead to cellular stresses that induce early senescence, DNA damage, and the extension of population doubling time [24,25,26]. Studies have shown that hypoxia preconditioning at levels 0.5–5% can upregulate various bioactive factors such as interleukin 6 (IL-6), TNF-α, HGF, and vascular endothelial growth factor (VEGF), resulting in enhanced cell proliferation and tissue regeneration in animal models after transplantation [27,28,29]. Moreover, hypoxia preconditioning has been observed to enhance the immunomodulatory capabilities of MSCs in various treatment scenarios [30,31,32,33].

Various priming methods, including inflammatory cytokines, hypoxia, 3D cultures, pharmacological or chemical agents, and biomaterials, have been discussed by Noronha et al. and Miceli et al. [7,34]. In this review, inflammatory cytokine priming (IFN-γ) and hypoxia were chosen specifically due to their consistent effects in boosting MSC immunomodulatory function and growth factors secretion. This review presents recent findings from pre-clinical studies on priming MSCs and their EVs with IFN-γ and hypoxia. It highlights the most commonly used priming concentrations and durations and therapeutic effects (the relationship between in vitro or functional markers with therapeutic effects). The review also discusses underlying mechanisms and the feasibility of scaling up these methods to meet industrial requirements.

During the preparation of this work, the authors used ChatGPT 3.5 for grammar and spelling checks. After using this tool/service, the authors reviewed and edited the content as needed and take full responsibility for the publication’s content.

## 2. MSCs Priming with IFN-γ

Numerous studies have demonstrated the effects of MSC priming with IFN-γ, a pro-inflammatory cytokine. Table 1 summarizes the diverse approaches reported in preclinical studies to achieve this priming, intending to boost MSCs’ immunomodulatory capabilities and enhance their secretion of immunosuppressive factors.

## 3. Phenotypic Characterization of Primed MSCs

Torkaman et al. observed no discernible alterations in the size, expansion, or morphology of the IFN-γ-primed human Wharton’s Jelly MSCs (hWJ-MSCs), which exhibited characteristics resembling fibroblast-like cells with star-shaped and spindle morphology [52]. Moreover, the presence of CD90, CD105, and CD73 markers was detected in both the primed and unprimed cells [43,52], while CD34, CD117, CD45, and CD31 markers were absent. This indicates that the isolated cells were mesenchymal stem cells, and the nature of hWJ-MSCs remained unchanged by IFN-γ, reaffirming that they retained their identity as mesenchymal stem cells [52]. Human Leukocyte Antigen-DR isotype (HLA-DR), CD80, CD86, and CD40 are pivotal in the development of immune responses, as well as the initiation and persistence of graft rejection issues [52]. Torkaman et al. showed that none of these surface molecules were expressed on primed and unprimed cells. However, Baudry et al. observed an increase in HLA-DR expression of the IFN-γ preconditioned human bone marrow-MSCs (BM-MSCs) [43,52]. Both IFN-γ-primed and unprimed MSCs exhibited adipogenic and osteogenic differentiation capabilities, indicating that priming did not alter the cells’ mesodermal characteristics [52,53].

## 4. IFN-γ Concentrations and Durations

Table 1 illustrates different IFN-γ priming concentrations and durations, influenced by factors such as MSC numbers, sources, culture protocols (e.g., mediums and serum supplements used and medium volume), and donor characteristics (age, health, and lifestyle). Bone marrow, umbilical cord and adipose tissues are the three main sources of MSCs in Table 1. However, no comparison was made between these sources. Nonetheless, all three MSC sources showed enhanced therapeutic effects upon the priming of different IFN-γ concentrations and durations. The use of U/mL as a concentration unit corresponds to IFN-γ specific activity but lacks a standardized method for determination. The concentrations of IFN-γ used are as low as 1 ng/mL and up to 50 ng/mL, whereas some studies used 500 U/mL and 200 IU/mL. It is noteworthy that the concentration units employed in these studies are not standardized, with some using ng/mL while others used U/mL or IU/mL. This inconsistency poses challenges for meaningful comparison between these study protocols. Rozier et al. demonstrated that extracellular vesicles (EVs) from MSCs pre-activated by a low dose of IFN-γ (1 ng/mL) were less efficient than naïve EVs, but when they increased the concentration to 20 ng/mL, improvement in several fibrotic, remodelling and inflammatory markers was observed [35]. In another study, Atsma lab showed that both unstimulated MSCs and 500 U/mL IFN-γ-stimulated MSCs have no significant beneficial effect on cardiac function or remodelling in a mouse model of myocardial infarction (MI) [53]. However, they previously reported that MSCs improved left ventricular function after acute myocardial infarction [54]. These discrepancies highlight the need for further research to establish optimal priming conditions for MSCs with IFN-γ, considering variations in experimental protocols and conditions.

Furthermore, the priming durations differ between 8 h and 7 days whereby the most common are 24 and 48 h. In an *Escherichia coli*-induced pneumonia rat model, EVs derived from MSCs primed using 50 ng/mL IFN-γ for 8 h are sufficient to increase the survival rate and decrease lung injury severity [44]. Nonetheless, Haan et al. showed no significant improvement even though the cells were primed using 500 U/mL IFN-γ for 7 days [53]. Studies proved that priming with IFN-γ upregulates IDO expression of MSCs in a dose- and time-dependent manner [41,55]; IDO is a key mediator of MSC immunomodulatory potency. However, prolonged exposure to IFN-γ can lead to MSCs’ immunosuppressive potency reverting to levels similar to unprimed MSCs [55].

TNF-α and IFN-γ-licensed human MSCs have been proven to possess a powerful anti-inflammatory effect in numerous studies [22,36,38,56]. Vaithilingam et al. demonstrated that stimulation with IFN-γ or TNF-α alone did not significantly induce the gene expression of chemokines (*C-X-C motif chemokine ligand 9* (*CXCL9*) and *CXCL10*) and the immunomodulatory cytokine (*interleukin-6; IL-6*) and *cyclooxygenase-2* (*COX-2*), while using a cytokine cocktail of IFN-γ and TNF-α induces synergistic effects on immunomodulatory properties of MSCs [50]. Another study compared the effect of different priming factors on IDO expression (TNF-α, IFN-γ, Lipopolysaccharide (LPS), Polyinosinic/polycytidylic acid (Poly I:C) alone or in combination) and revealed that higher concentration of IFN-γ + TNF-α (100 ng/mL each vs. 1 ng/mL) had the optimal effect on MSC expression of IDO [57].

In conclusion, the use of IFN-γ for priming MSCs is an active area of research, and there is still some variability in the methods used across studies. Despite this variability, the general consensus seems to be that priming MSCs with IFN-γ concentrations ranging between 10 ng/mL and 100 ng/mL for a duration of 24 to 48 h can enhance the therapeutic potential of MSCs. However, further validation is needed to establish a standardized priming protocol that can be consistently applied across different studies and clinical applications.

## 5. The Therapeutic Effects of IFN-γ-Primed MSCs (The Relationship between In Vitro or Functional Markers with Therapeutic Effects)

To investigate the therapeutic potential of primed MSCs, the expression levels of various cell markers such as IDO, PGE_2_, IL-10, IL-6, HGF, VEGF, TGF-β, matrix metalloproteinase-3 (MMP-3), CXCL9, CXCL10, C-C motif chemokine ligand 8 (CCL8), and others were measured following IFN-γ and/or TNF-α priming. Numerous studies have reported that increased IDO expression following IFN-γ exposure can be a key indicator of MSC immunomodulatory potency [38,39,40,41,45,49,50,55], while Kim et al. also confirmed that the elevated IDO expression in MSCs stimulated by IFN-γ is a widespread occurrence that was observed in all MSCs tested, regardless of their source. The immunosuppressive properties of IDO involve tryptophan degradation, which is essential for T-cell proliferation [49]. Takeshita et al. also found that in the presence of an IDO inhibitor, the inhibitory effects of IFN-γ-MSCs on PBMC proliferation were clearly diminished [45]. In addition to IDO expression, activation of MSCs by IFN-γ can also be confirmed by the upregulation of MHC II and programmed death-ligand 1 (PD-L1) expression on MSCs [39]. IFN-γ-primed MSCs could secrete significantly higher levels of immunomodulation factors, i.e., IL-10, HGF, VEGF and TGF-β than unprimed MSCs [52]. Additionally, the combination of IFN-γ and TNF-α stimulates MSCs to secrete a broad spectrum of pro-inflammatory and anti-inflammatory cytokines, including IL-4, IL-6, IL-10, and IL-13, promoting an anti-inflammatory phenotype [50]. The upregulation of cytokines such as CXCL9, CXCL10, and CCL8 by IFN-γ-primed MSCs may also contribute to the recruitment of leukocytes and various immune responses [49].

Numerous studies performed several functional tests to further correlate the effectiveness of IFN-γ priming including T-cell (PBMC) proliferation [8,39,45,49,52,53], Treg induction assay [39], bacterial phagocytosis and killing assays [44], macrophage polarity, motility, and phagocytosis [48], extracellular acidification rate (ECAR) [8,38], glucose uptake assay and hexokinase activity assay [38], as well as angiogenesis, proliferation, and migration assays (reparative or regenerative effects) [58,59,60]. Takeshita et al. proved that IFN-γ priming increased IDO expression and T cell suppressive properties in vitro, which were retained in a mouse calvarial defect xenograft of primed MSC extracellular matrix (ECM) complex (C-MSCs) [45]. The reduced neuroinflammation and improved functional outcomes in the EAE mice were correlated with the elevated IDO level, inhibited T-cell proliferation and induced Tregs after IFN-γ priming [39]. Additionally, Takeuchi et al. showed that IFN-γ-primed MSC-EVs contained anti-inflammatory macrophage inducible proteins, inducing anti-inflammatory macrophage responses in vitro and improving inflammation and fibrosis in a mouse model of cirrhosis [48].

Torkaman et al. reported that IFN-γ priming enhanced various immune suppression factors, including TGF-β, VEGF, HGF and IL-10, and significantly suppressed PBMC proliferation, leading to postponed clinical symptom onset in a mouse EAE model [52]. This highlights the multiple roles of growth factors/cytokines. Various growth factors, such as HGF, VEGF, TGF-β, and HIF-1α, have been shown to be upregulated after IFN-γ priming [58,59,60,61]. While VEGF and HGF are primarily associated with angiogenesis and proliferation, several studies have also indicated their involvement in the immunomodulatory properties of MSCs [19,52,62,63,64]. VEGF is believed to exert an immunosuppressive effect by inhibiting the migration or differentiation of bone marrow lymphoid precursors [64]. On the other hand, HGF induces IL-10 expression in monocytes, suppresses Th1 and dendritic cell activities, and promotes the generation of IL-10-positive regulatory T cells. Additionally, HGF produced by MSCs facilitates the expansion of immune-suppressive myeloid-derived suppressor cells (MDSCs) [63].

Notably, IFN-γ and TNF-α priming rapidly activate PI3K/AKT signalling and drive glycolysis in human MSCs, significantly reducing inflammatory parameters in mice with inflammatory bowel disease. This suggests that glycolysis affects the expression of immunosuppressive effectors such as IDO and TNF-stimulated gene-6 (TSG-6), as well as regulates the anti-inflammatory function of MSCs [38]. Conversely, Yao et al. found that IFN-γ priming shifts MSCs’ energy metabolism towards aerobic oxidation, activating the JAK/STAT pathway and inducing IDO and PD-L1 production. Combined ATP and IFN-γ treatment showed enhanced therapeutic efficacy, with ATP boosting the immunosuppressive capabilities of IFN-γ-primed MSCs via JAK/STAT pathway activation [8]. Metabolic programs profoundly affect immune cell responses; for instance, the rate of energetic consumption and biosynthesis of quiescent T cells is kept far lower than that of activated ones [65,66]. Upon stimulation, CD8^+^ T cells will rapidly switch to a metabolic profile characterized by accelerated rates of glucose uptake, glycolysis, and biosynthesis [67]. While the immunomodulatory functions of these immune cells are closely associated with distinct metabolic pathways, the metabolic requirements for the immunomodulatory function of MSCs remain to be fully clarified [38].

Despite their known enhanced immunomodulatory properties via paracrine effects, xenotransplantation of IFN-γ-primed human MSC extracellular matrix (ECM) complex (C-MSCγ) into an immunocompetent mouse calvarial bone defect model surprisingly induces bone regeneration, whereby lamellar bone was clearly observed in the lesion area [45]. This suggests that IFN-γ-primed MSCs can attenuate undesirable xenogeneic immune responses, facilitating successful bone regeneration. Moreover, in a notable study by Zhang et al., it was shown that treating MSCs with IFN-γ and TNF-α effectively eliminated donor-dependent variations in immunomodulation. Transcriptomic analyses revealed a positive correlation between MSCs’ immunomodulatory capabilities and the activation of IFN-γ and NF-κB signalling pathways induced by IFN-γ and TNF-α [22]. Additionally, MSCs primed with both IFN-γ and kynurenic acid (KYNA) exhibit significant therapeutic efficacy in addressing acute colitis and chronic colon fibrosis in rats via the induction of IDO-1. IDO-1 facilitated cell homing, polarization of intestinal macrophages to the anti-inflammatory M2 phenotype, and increased IL-10 expression to inhibit inflammation [40].

Other than that, numerous studies have harnessed the immunomodulatory property of MSCs for treating Type I diabetes, which is a well-known autoimmune disease characterized by specific adaptive immunity against β-cell antigens [50,68,69]. For instance, Wang et al. demonstrated that cytokine-primed MSC-EVs exhibited high levels of the immune checkpoint molecule PD-L1, which significantly reduced CD4^+^ T cell density and activation via the PD-L1/PD-1 axis and promoted the transition of macrophages from the M1 to M2 phenotype in mice with Type I diabetes [69]. Pancreatic islet transplantation is a therapeutic option for treating Type I diabetes; however, acute islet loss is a significant complication of this procedure. Barachini et al. reviewed various approaches using MSCs and MSC-EVs to create a more conducive immune microenvironment, aiming to reduce graft rejection and promote early vascularization to support graft survival [70]. Mrahleh et al. reported that MSCs primed with IFN-γ and TNF-α exhibited an immunomodulatory effect on CD4^+^ and CD8^+^ T cells by producing tolerogenic dendritic cells, which inhibit antigen-specific T cell responses via induction T cell anergy [68]. Similarly, Vaithilingam et al. reported improved encapsulated islet allograft survival and function via both co-encapsulation and co-transplantation of islets with primed MSC [50]. These findings have great implications for the future management and treatment of diabetes, which affects millions of patients worldwide [71].

In summary, while IFN-γ priming enhances therapeutic efficacy mainly via immunomodulatory properties and secretion of immunomodulatory factors, other reparative mechanisms also exist. Interestingly, studies show that IFN-γ and TNF-α priming induces glycolysis in MSCs, contributing to their anti-inflammatory properties [38]. In contrast, other studies have reported that IFN-γ alone induces aerobic respiration in MSCs, which is linked to their immunosuppressive ability [8]. Zhang et al. also demonstrated that treatment with IFN-γ and TNF-α can eradicate donor-dependent variations in MSC immunomodulation [22]. Additionally, primed MSCs have been shown to induce bone regeneration, although the mechanism remains unclear. IDO is an immunosuppressive enzyme that enhances the catabolism of tryptophan to kynurenine. Both the depletion of tryptophan and the accumulation of toxic kynurenine inhibit T cell proliferation and reduce neuroinflammation in EAE mice [72]. Therefore, the upregulation of MSC IDO expression and suppression of T-cell (PBMC) proliferation may serve as in vitro assays for assessing the immunomodulatory efficacy of IFN-γ priming in MSC and EV therapies. However, additional markers may also be evaluated to assess other therapeutic effects of interest that are concurrently influenced by IFN-γ priming.

## 6. Extracellular Vesicles Derived from IFN-γ-Primed MSCs

Comparative studies evaluating the influence of varying IFN-γ-preconditioning doses and durations on MSC-EV therapeutic efficacy are limited. One study compared the impact of different IFN-γ doses (1 ng/mL vs. 20 ng/mL) on EV therapeutic efficacy in systemic sclerosis in a murine model. They found that low-dose IFN-γ priming resulted in inferior suppression of dermal thickness and fibrosis by MSC-EVs compared to native EVs. However, primed EVs showed superior suppression of lung fibrosis and inflammation, partially ameliorating lung fibrosis. High-dose IFN-γ significantly enhanced EV-mediated amelioration of lung inflammation and fibrosis, with comparable effects on skin fibrosis suppression to native EVs [35]. This underscores the importance of optimizing priming protocols for achieving optimal outcomes. Figure 1 illustrates the changes in EV cargoes following IFN-γ priming.

Overall, preconditioning MSCs with IFN-γ, alone or combined with other factors, affects EV release, cargo composition, immunomodulatory activity, and therapeutic potential. For example, IFN-γ priming, either alone or with TNF-α, increases EV secretion from menstrual blood-derived and adipose tissue-derived MSCs or human umbilical MSCs, respectively [21]. Upregulated Rab27b and CD82 (involved in exosome secretion) may play a role in this observation [10]. However, no major changes were marked in EV size, size distribution [11,57,73], the expression of common EV surface and intravesicular markers [11,39], and protein concentration [44] in response to IFN-γ priming. Conversely, other studies reported that IFN-γ priming of bone marrow MSCs decreased EV secretion [74] or increased CD9 and CD81 EV surface protein levels in primed EVs [47]. Variances in the MSC tissue source, priming protocols, and MSC-EV separation techniques and storage conditions may account for these differences. Donor age may also influence MSC-EV secretion in response to priming. Cheng et al. demonstrated that pediatric MSCs produced more EVs and Rab27b than adult MSCs after IFN-γ + TNF-α stimulation [10].

**Figure 1 biomedicines-12-01369-f001:**
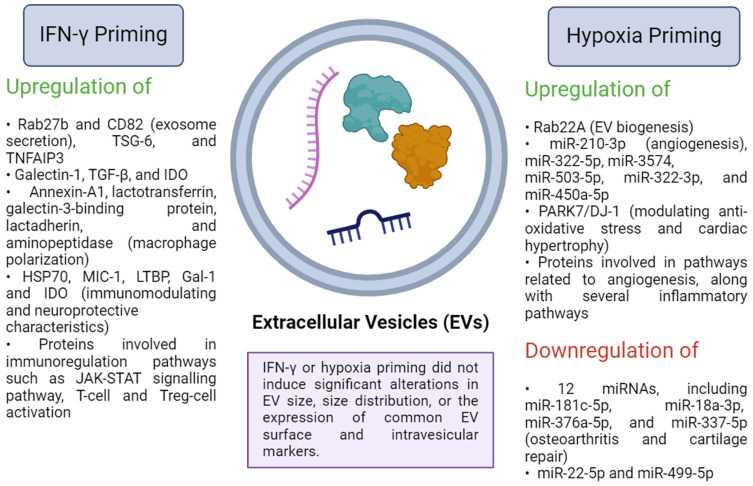
Alterations in EV cargo composition resulting from IFN-γ or hypoxia priming of MSCs [10,39,45,59,62,75,76,77,78,79,80]. The priming conditions influence the expression levels of various proteins, miRNA, and mRNA, ultimately enhancing the therapeutic potential of the primed EVs. IFN-γ, interferon-γ; TSG-6, tumour necrosis factor (TNF)-stimulated gene-6; TNFAIP3, tumour necrosis factor-alpha-induced protein 3; TGF-β, transforming growth factor-beta; IDO, indoleamine 2,3-dioxygenase; HSP70, heat shock protein 70; MIC-1, macrophage inhibitory cytokine 1; LTBP, latent transforming growth factor-binding protein; Gal-1, galectin-1; JAK, Janus kinase; STAT, signal transducer and activator of transcription; miR, miRNA; PARK7/DJ-1, Parkinson’s disease protein 7. Created with BioRender.com (accessed on 15 May 2024).

Profiling the proteomics of IFN-γ-primed MSC-EVs revealed a significant shift in EV expression of the immunomodulation-related proteins [10,11,39,48,81]. Cheng et al. reported that 183 proteins were exclusive to the primed adipose tissue-MSC-EVs (AT-MSC-EVs) as compared to the resting AT-MSC-EVs, including TSG-6 and tumour necrosis factor-alpha-induced protein 3 (TNFAIP3) (i.e., A20), which are crucial mediators of the immunomodulatory activity of MSCs. These primed AT-MSC-EVs suppressed activated ConA-activated T cells (CD4^+^), but AT-MSC-EVs inhibitory activity was significantly lower than that of their primed origin cells and their secretome [10]. Furthermore, Kim et al. identified a change in the proteomic cargo of IFN-γ-primed induced pluripotent stem cell-MSC (iPSC-MSC)-derived EVs, wherein 25 proteins were unique to the primed EVs, 101 protein expression significantly increased, and 181 protein expression significantly decreased. Additionally, primed EVs were enriched in proteins involved in immunoregulation pathways such as the JAK-STAT signalling pathway, T-cell and Treg-cell activation [81].

Serejo et al. found that IFN-γ preconditioning increased *galectin-1*, *TGF-β*, and *IDO* mRNA expression in adipose MSC-derived EVs but had no additional effect on EV immunoregulatory activity, as both IFN-primed and unprimed EVs inhibited T lymphocytes (in vitro) in a significant but comparable manner. Conversely, primed parent cells (adipose-MSCs) and their secretome showed a significantly greater T lymphocyte inhibition than the unprimed MSCs [73], implying that IFN-γ priming may have a varying effect on MSC and their derived EV functionality. Similarly, Takeuchi et al. detected a higher expression of proteins involved in macrophage polarization (i.e., annexin-A1, lactotransferrin, galectin-3-binding protein, lactadherin, and aminopeptidase) by adipose MSC-EVs in response to IFN-γ priming. Accordingly, treating macrophages with IFN-γ-primed EVs raised the expression of anti-inflammatory macrophage factors (IL-10, Ym-1, Fizz-1, CD206) and decreased the expression of pro-inflammatory factors (IL-6, TNF-α, monocyte chemoattractant protein-1 (MCP-1)) compared to the unprimed EVs, while both boosted macrophage motility and phagocytic activity in vitro [48].

Riazifar et al. showed that IFN-γ-MSC-EVs had a higher content of proteins with immunomodulating and neuroprotective characteristics, such as heat shock protein 70 (HSP70), macrophage inhibitory cytokine 1 (MIC-1), latent transforming growth factor -binding protein (LTBP) and galectin-1 (Gal-1). Profiling of RNA content revealed that, in comparison to resting MSC-EVs, IFN-γ-MSC-EVs contained an abundance of anti-inflammatory mRNAs such as *IDO*. Interestingly, both IFN-γ-MSC-EVs and unprimed MSC-EVs were found to be more enriched in non-coding RNAs (such as miRNA, lincRNA, and tRNA) than protein-coding RNAs relative to their parent cells, with the IFN-γ-MSC-EVs group showing significant enrichment of these noncoding RNAs compared with the unprimed MSC-EVs, implying their involvement in the regulation of several pathways. Interestingly, RNA functional inactivation by exposure to UV light led to a partial loss of MSC-EVs’ function [39].

On the other hand, studies that profiled the miRNA expression of IFN-γ-primed MSC EVs yielded contradictory results. While some studies demonstrated that priming had a limited effect on miRNA expression [35,48,82], others demonstrated a major change in miRNA, suggesting a significant role of modulated miRNA in the functionality of IFN-γ-primed EVs [21,39,47]. Interestingly, Rozier et al. reported that IFN-γ downregulated the majority of the modified miRNAs in MSCs. In contrast, no miRNA was found to be downregulated in IFN-γ-primed EVs. This study also revealed that IFN-γ -preconditioning differently altered the mRNA levels of several anti-inflammatory factors in MSCs and their generated EVs [35].

Furthermore, several studies demonstrated enhanced therapeutic potency of MSC-EVs with IFN-γ-priming in animal models of other conditions, including colitis [47], acute lung bacterial-induced pneumonia [44], cirrhosis [48], atopic dermatitis [81], and MI [21], highlighting the potential of IFN-γ-priming in enhancing MSC-EV therapeutic potency. For instance, treating the colitis mice model with IFN-γ-MSC-EVs considerably modified the condition indices and histological grade, along with reduced Th17:Treg cell ratios. In vitro, MSC-EVs inhibited Th17 differentiation, with IFN-γ-MSCs-EVs exhibiting a better inhibitory activity. This effect was suggested to be due to the upregulation of *miR-125a* and *miR-125b* in IFN-γ-primed EVs, which negatively regulated Stat3 and p-Stat3, leading to Th17 differentiation suppression [47].

To summarize, limited studies have explored the impact of IFN-γ pre-activation of MSCs on EV cargo and therapeutic efficacy, indicating a need for further research in this area. Evidence suggests that IFN-γ priming, alone or combined with other factors, enhances the expression of immunoregulatory factors in released EVs, potentially enhancing the therapeutic potential of MSC-EVs for immunological diseases. However, the effects of IFN-γ priming on EVs may vary depending on factors such as MSC tissue source, donor age, and priming protocol. While in vitro immunomodulatory effects of primed IFN-γ-EVs were inconsistent, better and more consistent outcomes were observed in vivo. Further research is necessary to understand the mechanisms underlying EV therapeutic functionality. Additionally, more studies are needed to compare the efficacy of primed MSCs and their derived EVs, as previous studies yielded conflicting results. Cargo analysis suggests potential distinct mechanisms of action between MSCs and EVs post-IFN-γ priming. Furthermore, the existing data suggest that IFN-γ-primed EVs act primarily on modulating immune cells such as macrophages [44,48] and T lymphocytes [39,47], an observation that warrants further investigation.

## 7. Long-Term Safety and Efficacy of IFN-γ Priming

In an equine model of osteoarthritis, Barrachina et al. conducted a 6-month monitoring study and noted that the beneficial effects of primed MSCs appeared to be short-lived. This may be attributed to the limited lifespan of MSCs following in vivo administration, particularly in cases of allogeneic and MHC-mismatched cells. Initially, both naïve and primed MSC treatments demonstrated an anti-inflammatory effect shortly after administration, especially after the initial injection when joint inflammation was more prominent. However, upon the second injection of allogeneic MSCs, a slight transient inflammatory reaction occurred, suggesting heightened immunogenicity of these cells. Notably, this reaction was observed only after the second injection, potentially indicating the development of immune memory, as evidenced by the recent discovery of functional antibodies against MHC-mismatched MSCs in horses [36]. In a GVHD mouse model, while all mice that were transplanted with hPBMCs alone or with a combination of hPBMCs and MSCs once had died, approximately 20% of mice that were transplanted twice with MSCs survived at 8 weeks post-transplantation. Notably, there seemed to be a trend suggesting a survival advantage among mice co-transplanted with hPBMCs and IFN-γ-primed MSCs compared to those receiving hPBMCs and naïve MSCs together. In this group, survival rates ranged between 40% and 60% at the 8-week post-transplantation mark [49]. Furthermore, in a study by Vaithilingam et al. in a diabetic mouse model, it was observed that all mice that received primed MSCs remained normoglycemic, in contrast to 71.4% of those in the unprimed MSC group at day 50 post-transplantation. Moreover, the viability of islets co-encapsulated with primed MSCs was significantly higher compared to those with unprimed MSCs at the same time point [50]. Additionally, Torkaman et al. demonstrated that neurologic functional recovery in EAE mice was significantly improved 50 days post-immunization with IFN-γ-primed MSCs compared to unprimed MSCs [52]. Safety monitoring in these studies was based on the premise that there were no manifestations of adverse health conditions. However, long-term systemic toxicity studies, including histopathological evaluation of vital organs and blood biochemical analysis, are lacking.

## 8. Signalling Pathways and Mechanisms of Action of IFN-γ Priming

Understanding the underlying mechanism of MSC therapy via the various signalling pathways is critical as it paves the way for a strong foundation for future scientific research and clinical applications for a variety of diseases. Moreover, each priming method exhibited distinct enrichment of molecular and cellular functions, which may explain their diverse mechanisms of action [37]. A schematic diagram of these pathways is illustrated in Figure 2.

Kim et al. demonstrated that IFN-γ induces IDO expression in MSCs via the Janus kinase/signal transducer and activator of transcription 1 (JAK/STAT1) signalling pathway, where both JAK1/2 and STAT1 in MSCs were activated following IFN-γ-stimulation [49]. IDO plays a direct role in the immunosuppressive properties of MSCs by suppressing antigen-driven T-cell proliferation. JAKs are associated with cytokine receptors, which are activated upon stimulation, and they phosphorylate STAT proteins, enabling them to be transported to the nucleus and thus regulate the transcription of target genes [83]. Nonetheless, activation of Toll-like receptor 3 (TLR3) in MSCs rarely induces IFN-β and/or IDO expression, suggesting that TLR signalling is not a major pathway in MSC immunosuppressive functions [49]. Moreover, Yao et al. also showed that IFN-γ-induced metabolic reconfiguration to aerobic oxidation activates the JAK/STAT pathway, which is necessary for expressing IDO and PD-L1 in MSCs [8]. Similarly, Ling et al. also reported that IFN-γ-MSCs could reduce inflammation in experimental autoimmune encephalomyelitis (EAE) mice via the forkhead box P3 (Foxp3)/retinoic acid-related orphan receptor gamma t (ROR-γt)/STAT3 signalling pathway, whereby this is the downstream pathway of JAK/STAT signalling [42]. This pathway involves Foxp3 directly interacting with ROR-γt and STAT3 signalling, crucial for inducing ROR-γt and subsequent T helper 17 (Th17) cell differentiation [84]. The imbalance of Th17/Tregs is implicated in the pathogenesis of EAE [42].

Analysis of global mRNA profiles from MSCs primed with poly I:C or IFN-γ revealed enrichment of canonical pathways associated with cell survival and inflammatory responses, including interferon signalling and the antigen-presenting pathway, compared to non-primed MSCs. At the same time, IFN-γ priming has been shown to affect much more complex functions, including cellular development, cell growth and proliferation, cell-to-cell signalling and interactions, cell-mediated immune responses and cell movement [37]. Moreover, Riazifar et al. used an ingenuity pathway analysis database of gene ontology and demonstrated that the phosphoinositide 3-kinase/Ak strain transforming (PI3K/AKT) signalling pathway is the most relevant canonical pathway affected by miRNAs enriched in IFN-γ-primed exosomes, as 19 out of 123 genes in this pathway are affected [39]. Similarly, Xu et al. also showed that TNF-α and IFN-γ acutely activate the PI3K/AKT signalling pathway, essential for the expression of IDO and TSG-6. TSG-6, secreted from stimulated MSCs, binds to hyaluronan and reduces inflammation in various disease models [38]. The activation of the PI3K/Akt pathway begins with IFN-γ binds to its receptor and then activates JAK2 to phosphorylate STAT1, where phosphorylated STAT1 further activates PI3K and continues with the recruitment and activation of the inactive signalling protein AKT [75].

Numerous studies have indicated that IFN-γ-primed MSCs suppress T cell proliferation, regulate the activity of helper T cells (Th), and induce regulatory T cells (Tregs), suggesting that T cells play a critical role in the immunomodulatory properties of primed MSCs [37,39,42,45,47,48,49,51,52]. Yang et al. showed that increased levels of *miR-125a* and *miR-125b* in primed MSC-EVs inhibit Th17 cell differentiation, leading to increased efficacy in treating colitis [47]. Additionally, the upregulation of anti-inflammatory cytokines IL-4, IL-6, IL-10 and granulocyte colony-stimulating factor, as well as enhanced nitric oxide production, help modulate the immune response and improve the immunosuppressive capacity of MSCs [50]. Baudry et al. reported that increased red blood cell velocity, rolling white blood cell flux and number of venules with circulating white blood cells, as well as reduced rate of soluble E-selectin (which may serve as a biomarker to monitor an organ’s endothelium damage), can improve microvascular hemodynamics in early stages of sepsis [43].

In an EAE mouse model, the IFN-γ-primed exosomes were not detected after a short period in vivo, suggesting a “hit and run” mechanism for their long-lasting efficacy, likely via immunomodulatory mechanisms such as induction of Tregs for immune tolerance [39]. Furthermore, Park et al. showed that IFN-γ-primed MSCs regulate *Aspergillus fumigatus*-induced immune responses via the regulation of Th17 immune responses, whereas poly I:C-primed MSCs control both eosinophil-associated Th2 immunity and neutrophil-related Th17 immunity [37]. Other than that, Varkouhi et al. reported that the beneficial effects of MSC-EVs appear to be driven by the enhancement of macrophage phagocytosis and killing of *Escherichia coli* [44]. In an inflammatory bowel disease mouse model, primed MSCs reduced inflammatory parameters via enhanced IDO and TSG-6 expression, promoting glycolysis, glucose uptake, and hexokinase II activity [38]. Additionally, Ye et al. showed that IFN-γ and KYNA combination induces IDO-1 expression, facilitating cell homing, M2 polarization of intestinal macrophages, and increased IL-10 expression, effectively inhibiting inflammation in acute colitis and chronic colon fibrosis [40].

In summary, research on IFN-γ priming of MSCs elucidated its mechanisms and therapeutic implications across various diseases. IFN-γ triggers IDO expression in MSCs via the JAK/STAT1 pathway, facilitating immunosuppression and T-cell regulation. Moreover, the PI3K/AKT pathway plays a crucial role in mediating IDO and TSG-6 expression. IFN-γ-primed MSCs exhibit diverse effects, impacting cellular functions and canonical pathways associated with survival and inflammation. Notably, IFN-γ priming influences T cell activities, promotes Treg induction, and modulates cytokine production. Furthermore, the “hit and run” mechanism of IFN-γ-primed extracellular vesicles underscores their long-lasting immunomodulatory effects.

## 9. MSCs Priming with Hypoxia

Several studies have demonstrated the effects of conditioning MSCs in hypoxic environments, contrasting with standard cell culture conditions of 20–21% oxygen levels. Table 2 outlines key findings from preclinical studies focusing on priming MSCs in reduced oxygen environments. This method aims to enhance MSC proliferation and longevity while also amplifying their angiogenic and immunomodulatory capacities.

## 10. Phenotypic Characterization of Primed MSCs

Wu et al. found that long-term hypoxic treatment did not alter the stem cell characteristics of human UC-MSCs, including morphology, surface biomarker expressions, differentiation ability, tumorigenicity, and chromosomal stability [30]. Similarly, Kim et al. demonstrated that the expression of MSC surface marker proteins (CD73, CD90, CD105, and CD166) and multipotency remained comparable between small MSCs primed with hypoxia and calcium ions (SHC-MSCs) and naïve MSCs [33], suggesting that hypoxia preconditioning did not induce significant alterations in MSC characteristics. However, in contrast, Hu et al. reported that mouse BM-MSCs exposed to 5% oxygen exhibited a notably more flattened spindle-shaped morphology, less convex compared to those in 21% oxygen [88].

## 11. Hypoxia Priming Methods (Oxygen Levels) and Durations

Oxygen levels in these studies ranged from 0.5% to 5%, with 1% and 5% being the most common. Additionally, 1% oxygen tension has been frequently used in acute kidney injury (AKI) and renal ischemia reperfusion injury (IRI) disease models [76,77,78], whereas 5% oxygen tension has been frequently used in diseases related to immunomodulation which included allotransplantation, traumatic brain injury (TBI), skin–wound healing and chronic asthma [32,79,80,87]; yet there is no direct relationship observed between oxygen tension and disease models. Moreover, the durations of priming in these studies typically range from 24 h to 72 h. However, Kim et al. kept the MSCs under hypoxic conditions (3% O_2_) in media supplemented with calcium ions throughout the entire culturing process [33], similar to Wu et al., who cultured the MSCs under 1% O_2_ throughout the entire culturing process [30]. Additionally, the three primary sources of MSCs listed in Table 2 are the bone marrow, umbilical cord, and adipose tissues. While no direct comparison was conducted, all three sources exhibited improved therapeutic effects when subjected to priming with different oxygen levels and durations.

Interestingly, Kim et al. utilized small MSCs primed with hypoxia (3% O_2_) and 1.8 mM calcium ions (SHC-MSCs), isolated by filtering MSCs through a pluriStrainer with a pore size of 10 μm. Small MSCs were selected because they are readily self-renew whereas large cells tend to lose this characteristic, and they are less likely to become trapped in capillaries, which would be beneficial for its clinical application. They reported that SHC-MSCs have an enhanced potency for treating GVHD as compared to naïve MSCs [33]. Moreover, Ishiuchi et al. demonstrated that serum-free medium and hypoxia preconditioning (1% O_2_) synergistically enhanced the proliferative capacity and anti-fibrotic effects of MSCs [77]. These results suggested that serum-free conditions and hypoxic preconditioning do not antagonize each other but enhance paracrine activity. Hypoxia preconditioning improved the direct anti-fibrotic effect of serum-free MSCs but did not enhance their anti-inflammatory effect, likely due to pre-existing strong anti-inflammatory effects of serum-free MSCs. Surprisingly, Soares et al. compared priming MSCs with hypoxic condition (5% O_2_) or 1000 units of IFN-γ for 72 h and found that hypoxic priming is significantly better than IFN-γ priming in prolonging allograft rejection [79]. One of the possible explanations is that IFN-γ priming will convert MSCs to an antigen-presenting cell phenotype, thereby decreasing their immunomodulatory capability [89].

In summary, the most commonly used hypoxic conditions and durations were 1% oxygen level and 24 h, respectively. Yet there is no standard protocol for this hypoxia preconditioning, and hence, more studies can be carried out to compare the effect of different hypoxic conditions and durations on enhancing MSC therapies.

## 12. Therapeutic Effects of Hypoxia-Primed MSCs (The Relationship between In Vitro or Functional Markers with Therapeutic Effects)

To investigate the therapeutic efficacy of MSCs after hypoxia priming, the expression level of a few cell markers was measured, which include hypoxia-inducible factor (HIF)-1, leptin, basic fibroblast growth factor (bFGF), VEGF, IDO, HGF, PGE_2_ and more. HIF-1, a dimeric protein complex crucial for the body’s response to low oxygen levels, plays a key role in vascularization in hypoxic areas like localized ischemia and tumours [90]. It serves as the upstream regulator of several key genes, including VEGF, HGF, PGE_2_, and IDO [91]. Studies have consistently shown the induction of hypoxia-inducible factor-alpha (HIF-1α) expression in MSC cultures under hypoxic conditions [92,93]. Additionally, hypoxic MSCs exhibit enhanced levels of angiogenic factors such as VEGF and bFGF, along with improved antioxidative capacity. Paracrine effects seem to be the main mechanism by which hypoxic MSCs modify kidney effects of ischemia-reperfusion injury (IRI) [78]. Similarly, other studies also showed that hypoxia-preconditioned MSCs enhance the production of bFGF, VEGF, HGF and PGE_2_ [76,77,87]. Knockdown experiments suggest that VEGF acts as an upstream effector of HGF, and both VEGF and HGF knockdown diminish the anti-fibrotic effect of hypoxia-preconditioned MSCs [76,77].

Numerous studies have conducted various functional tests to further establish the effectiveness of hypoxia priming. These include MSC migration assays [77,79,85], MSC apoptosis assays [85], tube formation assays [33,85], assessments of human T-cell (PBMC) proliferation [33,79], macrophage polarization assays [77], measurements of oxygen consumption rate (OCR), mitochondrial membrane potential, reactive oxygen species (ROS) levels [86] and more. Hypoxia preconditioning induces CXCR4 expression in a leptin-dependent manner, enhancing MSC homing, survival, and angiogenesis, ultimately improving cardiac function in a mouse model of myocardial infarction (MI) [85]. Intriguingly, hypoxia priming upregulates IDO expression, inhibiting the proliferation of CD4^+^ T cells and boosting the motility and proliferative potential of MSCs. This, in turn, delays the onset of acute rejection while preserving the recipient’s adaptive immune response [79]. Surprisingly, hypoxia priming also enhances the expression of Parkinson’s disease protein 7 (PARK7/DJ-1) in its EVs, resulting in the alleviation of mitochondrial damage and the suppression of angiotensin II type 1 receptor (AT1R)-associated protein (ATRAP) degradation [86]. Prolonged hypoxia exposure improves MSC proliferative capacity and telomerase activities, reduces senescence, and preserves multipotency compared to normoxic conditions. Hypoxic EVs inhibit immune activation via VEGF-mediated inhibition of dendritic cell maturation and downregulation of costimulatory molecules and HLA-DR expressions [30].

Interestingly, priming MSCs in hypoxic environments (5% O_2_) before IRI resulted in a 4.3-fold increase in *IDO* transcript expression compared with normoxic MSCs (21% O_2_). The addition of a competitive IDO inhibitor to the hypoxia-primed MSCs, unexpectedly, did not significantly affect allograft survival, while hypoxia-primed MSCs are better than IFN-γ priming at preventing allograft rejection. This suggests that hypoxic conditions may drive IDO expression to a level where the competitive inhibitor no longer effectively saturates enzymatic binding sites [79]. Remarkably, Hendrawan et al. demonstrated that conditioned medium (CM) from hypoxic MSCs facilitated wound repair more effectively in early-stage diabetic wound models compared to antibiotic treatment commonly used for diabetic foot ulcer care [87]. However, in an atherosclerotic renal artery stenosis (ARAS) porcine model, Farooqui et al. reported that hypoxia preconditioning of MSCs showed comparable effects to normoxia and did not enhance the therapeutic impact of MSCs on renal function, despite inducing a reduction in DNA hydroxymethylation (5hmC levels) of inflammatory and profibrotic genes [31].

Additionally, Nowak-Stȩpniowska et al. found that 19 out of 30 conditions (varying in oxygen concentration and incubation time) led to increased MSC proliferation in oxygen concentrations ranging from 1 to 5%. The higher proliferation in hypoxic conditions may result from the shift to anaerobic respiration, leading to increased glucose consumption and lactate generation in MSCs [94]. Hypoxia also diminishes cellular ATP consumption and the production of ROS, hence preventing bioenergetic collapse [95]. Intriguingly, Xu et al. reported a non-invasive approach involving nebulized hypoxic EV inhalation, significantly reducing chronic airway inflammation and remodelling. The EVs predominantly accumulate in the lungs and persist for a duration of 7 days [32].

In conclusion, HIF-1 expression serves as an indicator of the effectiveness of hypoxia priming in MSC therapies. Hypoxia priming enhances MSC properties including angiogenesis, anti-fibrotic effects, anti-apoptosis, and cell homing while also suppressing inflammatory cell infiltration via IDO expression and improving MSC antioxidative capacity. Additionally, hypoxia-primed MSCs exhibit a shift from aerobic to anaerobic respiration and increased glucose uptake.

## 13. Extracellular Vesicles Derived from Hypoxia-Primed MSCs

There is a scarcity of comparative studies assessing the impact of varying oxygen levels and durations on the therapeutic efficacy of MSC-EVs. In general, hypoxic EVs have demonstrated better therapeutic efficacy compared to normoxic EVs [30,86]. Wu et al. reported that both hypoxic EVs and conditioned medium (CM) exhibit more pronounced therapeutic effects than their normoxic counterparts, suggesting that the essential components in the CM likely originate from EVs [30]. Studies have indicated that there is no significant difference in average size and marker protein expression between normoxic EVs and hypoxic EVs [30,86,96], revealing that hypoxia priming does not induce significant alterations in the characteristics of MSC-EVs. Interestingly, Xu et al. designed an inhalation device for asthma mice, showing that nebulized hypoxic EVs and hypoxic EVs have similar round nanoparticle structures and complete membranous integrity, indicating maintained structural integrity of nebulized hypoxic EVs [32].

There are indications that the induction of HIFs could directly impact the pathways of EV biogenesis involving Rabs [97]. In tumour B cells, HIFs were shown to directly bind to the *Rab22A* locus, leading to the expression of Rab22A, a protein necessary for the budding of microvesicles from the plasma membrane. Silencing *Rab22A* expression prevented hypoxia-induced EV release [98]. Andrew et al. demonstrated that hypoxic priming increased EV release, while inflammatory priming (TNF-α and IFN-γ) affected EV size [99]. Similarly, in a review paper, Bister et al. reported that quantitative measurements of EVs provided evidence of increased EV release after hypoxia compared to normoxia. However, the exact mechanisms controlling the fate of multivesicular bodies (MVBs) or the release of EVs in response to hypoxia in other cellular contexts still need clarification [97].

In an osteoarthritis rat model, Zhang et al. reported that miRNA sequencing indicated changes in miRNA expression due to hypoxia pretreatment. Specifically, 12 miRNAs, including *miR-181c-5p*, *miR-18a-3p*, *miR-376a-5p*, and *miR-337-5p*, showed significant alterations. Hypoxia priming was demonstrated to decrease the expression of these miRNAs, and their target genes, such as *ALB*, *STAT3*, and *MAPK1*, are associated with osteoarthritis and cartilage repair [96]. Furthermore, Zhuang et al. illustrated that the expression of *miR-210-3p*, *miR-322-5p*, *miR-3574*, *miR-503-5p*, *miR-322-3p*, and *miR-450a-5p* significantly increased in hypoxic small EVs compared to normoxic small EVs, while the expression of *miR-22-5p* and *miR-499-5p* was inhibited. Importantly, previous evidence has suggested that *miR-210* may play a role in the angiogenic process in various cell types, and it is known to be hypoxia-responsive, consistent with the sequencing results [100].

Furthermore, Lu et al. conducted a quantitative proteomics analysis to discern differential protein expressions between normoxic EVs and hypoxic EVs. They identified PARK7/DJ-1 as the protein with the most significant expression difference, with higher levels in hypoxic EVs. DJ-1 plays a role in modulating anti-oxidative stress and cardiac hypertrophy [86]. Additionally, Xu et al. observed that hypoxic EVs exhibited greater protein diversity compared to normoxic EVs, identifying a total of 395 proteins, including 74 unique ones. Pathway analysis of these unique proteins revealed enrichment in pathways related to angiogenesis, such as “cell migration” and “cell adhesion”, along with several inflammatory pathways. Intriguingly, they observed reduced EV secretion levels when cells were exposed to both nutritional starvation and oxygen-depleted conditions [101]. The changes in EV cargoes following hypoxia priming were also illustrated in Figure 1.

In summary, limited studies have delved into the impact of diverse oxygen levels and durations on MSCs concerning EV cargo and therapeutic effectiveness, indicating a necessity for further investigation in this realm. The existing evidence suggests that hypoxia priming, involving variations in oxygen levels and durations, enhances the expression of therapeutic factors in released EVs, presenting a potential avenue for amplifying the therapeutic efficacy of MSC-EVs. Notably, the influence of hypoxia priming on EVs may exhibit variations contingent on factors such as MSC tissue source, donor age, and the priming protocol employed. Additional studies are imperative for a comprehensive understanding of the mechanisms underpinning the therapeutic functionality of EVs. Furthermore, cargo analysis of both MSCs and EVs supports the likelihood of distinct mechanisms of action, as the impact of hypoxia priming on the cargo of MSCs and EVs may differ.

## 14. Long-Term Safety and Efficacy of Hypoxia Priming

In a humanized mouse model of GVHD, the administration of SHC-MSCs led to improved survival, reduced weight loss, and decreased histological evidence of GVHD 6 weeks post-transplantation. Eight weeks after transplantation, approximately 90% of mice transplanted with human PBMNCs alone (GVHD group) had died, whereas the majority of mice transplanted with naïve MSCs (80%), SHC-MSCs (90%), and Polo-like kinase-1 (PLK1)-overexpressing MSCs (90%) survived. In line with the anti-inflammatory and immunomodulatory activities observed in vitro, the levels of human inflammatory cytokines, including IL-2, TNF-α, and IFN-γ, were more effectively reduced in the blood samples of GVHD mice treated with SHC-MSCs compared to those treated with naïve MSCs [33]. Long-term safety has not been established in the study.

## 15. Signalling Pathways and Mechanisms of Action of Hypoxia Priming

In an MI mouse model, Hu et al. showed that leptin signalling is a critical initial step in the enhanced survival, chemotaxis, and therapeutic capabilities of MSCs imparted by hypoxia preculture [85]. HIF-1α is a molecular sensor of hypoxia, and it transactivates the leptin gene expression by HIF-1α transcription-dependent activity. Leptin is a product of a hypoxia-inducible gene, and hypoxia differentiates the preadipocytes into leptin-secreting endocrine cells in an mTOR-dependent manner [102]. Leptin, a multifunctional cytokine primarily produced by adipocytes, regulates various processes, including appetite control, energy balance, metabolism, cell survival, migration, and angiogenesis [103,104,105]. Upon binding to its transmembrane receptors (ObR), leptin triggers the autophosphorylation of two JAK molecules, leading to the activation of STAT proteins via tyrosine phosphorylation [106]. Hence, leptin can also activate STAT-independent pathways, including extracellular signal-regulated kinase (ERK) and PI3K cascades [107]. Moreover, the possible mechanisms involved in this leptin signalling are STAT3/HIF-1α/VEGF signalling and stromal cell-derived factor 1 (SDF-1)/C-X-C chemokine receptor type 4 (CXCR4) signalling that leads to enhanced MSC engraftment, and cardiac protection via autocrine and paracrine effects as well as possibly recruitment of endogenous progenitor cells. Their findings support the hypothesis that STAT3 is part of the leptin signal pathway and plays an important role in enhanced cardio-protection by hypoxia priming. STAT3 enhances the HIF-1α actions by binding and stabilizing the transcription factor complex, and this is required for sufficient activation of several hypoxia-activated genes, including *VEGF* [108], while SDF-1/CXCR4 signalling promotes cell homing and survival where the induction of CXCR4 expression is dependent, at least in part, on HIF-1α [109].

Other than that, Ishiuchi et al. found that MSC-conditioned medium inhibits TGF-β/Suppressor of Mothers Against Decapentaplegic (Smad) signalling as a mechanism by which MSCs exert their anti-fibrotic effect. The conditioned medium from 1% O_2_ (hypoxia) MSCs contained HIF-1α that further suppressed TGF-β1-induced phosphorylation of Smad2 and alpha-smooth muscle actin (α-SMA) compared with 21% O_2_ (normoxic) MSCs [76]. Numerous studies demonstrated that TGF-β is the key mediator in chronic kidney diseases associated with progressive renal fibrosis, where TGF-β1 will activate Smad2 and Smad3 that bind directly to several microRNAs to either negatively or positively regulate their expression and function in renal fibrosis [110]. Similarly, subsequent studies found that hypoxia and serum-free medium-cultured MSCs (hypo-SF-MSCs) also inhibit TGF-β/Smad signalling, with HGF knockdown attenuating this effect [77]. A schematic diagram of these pathways is illustrated in Figure 2.

Fascinatingly, Lu et al. documented that PARK7/DJ-1 (Parkinson’s disease protein 7) derived from hypoxic EVs mitigates mitochondrial damage and inhibits proteasome subunit beta type 10 (PSMB10) activity via direct interaction. This reduces AT1R-associated protein (ATRAP) degradation, thus inhibiting AT1R-mediated signalling and mitigating cardiac hypertrophy [86]. DJ-1 is vital for maintaining mitochondrial function and acting as an endogenous antioxidant, offering cardiovascular protection [111,112]. In a mouse model of allergic rhinitis (AR), extended exposure to hypoxia has been observed to elevate VEGF levels in EVs. This increase, in turn, suppresses the differentiation and maturation of dendritic cells, which are pivotal in the pathogenesis of AR [30]. Intriguingly, Soares et al. illustrated that hypoxia priming, similar to IFN-γ priming, enhances IDO expression, protecting endothelial cells, inhibiting CD4^+^ T cell proliferation, and boosting motility and proliferation potential [79].

In conclusion, leptin signalling plays a crucial role in enhancing the survival, chemotaxis, and therapeutic capabilities of MSCs induced by hypoxia preconditioning. Leptin, primarily produced by adipocytes, activates various pathways, including JAK/STAT, ERK, and PI3K cascades upon binding to its receptors. Moreover, an MSC-conditioned medium has been shown to inhibit TGF-β/Smad signalling, further suppressed under hypoxic conditions, offering anti-fibrotic effects. PARK7/DJ-1 from hypoxic EVs mitigates mitochondrial damage and inhibits PSMB10 activity, reducing ATRAP degradation, thereby inhibiting AT1R-mediated signalling and mitigating cardiac hypertrophy.

## 16. Primed MSC in Clinical Trials

Establishing standardized and consistent markers for evaluating the efficacy of IFN-γ and hypoxia priming is essential for ensuring reproducible and reliable results across different studies. Key potency markers such as IDO, PGE2, VEGF, TGF-β, and HIF-1 should be incorporated into the product validation protocol, along with functional assays like activated T cell inhibition before clinical use. Consistency in marker selection is vital for meaningful comparison of results between studies, facilitating a clearer understanding of the priming mechanisms and optimal conditions for MSC-based therapies. It is also crucial to note that while some studies have shown positive results with IFN-γ and hypoxia priming, others have reported limited or no improvement when compared to unprimed MSCs [31,35,46,51,53]. This highlights the importance of further research to fully understand the multiple mechanisms at play and determine the conditions under which IFN-γ and hypoxia priming provide the best therapeutic outcomes. In addition, IFN-γ priming can be combined with other priming strategies, such as hypoxia. For example, an in vitro study reported that hypoxia is a relatively low-cost addition to IFN-γ priming that leads to additive immunosuppressive effects over IFN-γ priming alone [113].

According to the ClinicalTrials.gov database, there are two ongoing phase 1 clinical trials using IFN-γ-primed MSCs for xerostomia following radiotherapy and asthma started in the year 2022. Both trials employ IFN-γ-primed MSCs derived from human bone marrow, although the specific method and duration of IFN-γ priming have not been revealed. The investigators claimed that the risk of cell therapy using IFN-γ-primed MSCs is comparable to that of naïve MSCs, which possess an outstanding and well-established safety profile [114]. The Good Manufacturing Practice (GMP) protocol for IFN-γ-primed MSCs has been developed and demonstrated to be safe, feasible, and reliable in pre-clinical models. The upscaling of the production of IFN-γ-primed MSCs has also been shown to be feasible in this GMP setting. The two-step approach involves MSC isolation, expansion and cryopreservation, followed by thawing and priming with IFN-γ for 7 days before transplantation [114]. Such an approach would pose logistical challenges, including 1. a longer lead time for product requests as the product is not on the shelf; 2. The risk of unsuccessful or insufficient priming, which could delay product release; and 3. the need to study and validate the stability of primed MSCs before considering the transfer of the products to off-site clinical facilities. It will be interesting to investigate if IFN-γ-primed MSCs can withstand cryopreservation and retain their immunomodulatory properties. If this is shown to be feasible, it will greatly ease the logistic issue of scheduling transplantation and reduce the cost of manufacturing.

Other than that, there are two ongoing phase 2 clinical trials initiated in 2020 and 2022 utilizing hypoxia-primed MSCs for severe COVID-19 and chronic lumbar disc disease (cLDD), respectively. The COVID-19 trial employs secretome from hypoxic MSCs, while the cLDD trial utilizes autologous bone marrow hypoxic MSCs. In a chronic lower back pain case report, researchers observed that intra-discal injection of autologous, hypoxically cultured bone marrow-derived MSCs demonstrated safety and feasibility in five patients diagnosed with degenerative disc disease. The BM-MSCs were cultured in 5% oxygen throughout the entire culturation process, and patients were followed up 4–6 years post-MSC infusion [115]. Additionally, Putra et al. demonstrated that hypoxia-preconditioned MSCs and the secretome exhibit superior effects in treating various diseases, including acute renal failure [116], full-thickness-wound [117], and more. While these ongoing trials suggest promise for the priming strategy, its effectiveness in humans awaits the publication of clinical trial data. The specific method and duration of hypoxia priming were not disclosed in these trials, but they indicate the feasibility of upscaling hypoxic MSC production.

For MSC up-scaling, various bioreactors have been developed. Due to the anchorage nature of these cells, most bioreactors require the use of multiple culture containers in stacks (cell factories) or the use of cell microcarriers, scaffolds or hollow fibres to increase the surface area for cell attachment and proliferation [118,119,120]. The GMP production of hypoxic MSCs would likely require the entire cell manufacturing process to be performed in hypoxic workstations, a solution already commercially available and expected to become a mainstay for GMP cell manufacturing facilities. Newer GMP facilities should adopt modular installations where a flexible layout and custom-made workstations can be accommodated. This allows change to be implemented easily as part of the evolving needs of the industry. The adoption of the two-step approach, as mentioned above, may work as well for hypoxia priming, i.e., firstly, the large-scale expansion of MSC followed by cryopreservation, and secondly, the priming of revived MSCs in a hypoxic chamber. Such an approach faces similar challenges as previously mentioned.

EVs are isolated from the conditioned medium of MSC cultures, with isolation techniques varying between laboratories. Solutions for handling and processing large volumes of liquid are already established in the pharmaceutical industry. Tangential flow filtration may be preferred due to its gentle processing [121]. Regardless of the methods used, the stability and storage of EVs must be evaluated. Given the short half-life of EVs, freeze-drying using cryoprotectants like mannitol has been developed [122].

## 17. Ethical and Regulatory Considerations

Ethical concerns regarding MSCs and their EVs have largely been addressed by using discarded tissues such as umbilical cords. However, it is crucial to obtain written informed consent, ensure transparency about the use and handling of donated samples, and protect donor confidentiality at all times. The European Medicines Agency classifies MSCs and their EVs as Advanced Therapeutic Medicinal Products (ATMPs) (Advanced therapy medicinal products, https://www.ema.europa.eu/en/human-regulatory-overview/advanced-therapy-medicinal-products-overview (accessed on 17 May 2024)), while the US Food and Drug Administration categorizes them as Cell & Gene Therapy Products (CGTPs) (Framework for the Regulation of Regenerative Medicine Products, https://www.fda.gov/vaccines-blood-biologics/cellular-gene-therapy-products/framework-regulation-regenerative-medicine-products (accessed on 17 May 2024)). While jurisdiction in emerging countries may still be in its infancy, most developing regulators will adopt the guidelines from these mature agencies. The primary concern in the regulation of medicinal products is safety besides their purity and quality. In general, the safety of these products after extensive expansion and manipulation must be assessed. In addition to incorporating a potency or functional assay, there is also a need to rule out tumorigenicity and abnormal karyotype for the MSCs. To minimize off-target effects, it is recommended to measure known markers for side effects, such as apoptotic or senescent markers and inflammatory markers, in MSCs and their EVs as exclusion release criteria. Such safety concerns may be heightened after cell priming due to the pre-stimulated state and increased potency of MSCs or their EVs. Since IFN-γ is a potent inducer of immune cell proliferation, the safety of its residual amount in primed MSCs and in the EV cargo that is administered should be established. A cut-off level for the residual IFN-γ and in the EV cargo needs to be ascertained before release. Additionally, it is essential to source clinical and GMP-grade IFN to ensure quality and consistency for priming.

## 18. Conclusions and Future Recommendations

In conclusion, the current review highlights the potential of IFN-γ and hypoxia priming, either singularly or in combination, to enhance the therapeutic effectiveness of MSCs in regenerative medicine. This enhancement occurs via the modulation of signalling pathways such as JAK/STAT and PI3K/AKT, and Leptin/JAK/STAT and TGF-β/Smad. IFN-γ and hypoxia priming of MSCs influence their extracellular vesicle cargo and show promise as a cell-free therapeutic approach. The upregulation of IDO expression, which further suppresses T-cell proliferation upon IFN-γ priming, along with the consistent expression of HIF-1 after hypoxia priming, suggest their potential as reliable in vitro markers for evaluating the efficacy of these priming methods. Therefore, we propose considering them as tests for developing a standardized protocol (Figure 3). However, no specific methodologies have yet demonstrated promising repeatability in EVs derived from primed MSCs. Presumably, the therapeutic effects of primed MSCs will be passed down to their EVs. However, quality control should be performed both at the EV cell source and the EVs.

Despite significant advancements, several knowledge gaps persist. Currently, standardization of priming protocols concerning IFN-γ and hypoxia is lacking, highlighting the need for further research to establish optimal conditions. Investigation into the metabolic profiles of IFN-γ and hypoxia-primed MSCs is recommended to provide insight into the best optimization strategies and improve our understanding of the underlying mechanisms.

## Figures and Tables

**Figure 2 biomedicines-12-01369-f002:**
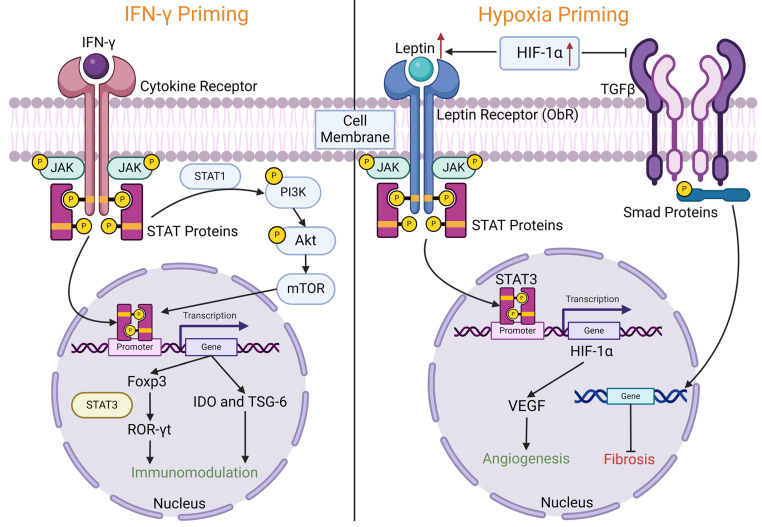
Signalling pathways underlying MSCs priming with IFN-γ or hypoxia. IFN-γ priming activates the JAK/STAT and PI3K/AKT pathways, eventually enhancing the immunomodulation properties of MSCs upon priming. In contrast, hypoxia priming activates the Leptin/JAK/STAT pathway and inhibits the TGF-β/Smad pathway, thus empowering the angiogenesis and anti-fibrotic properties of MSCs after priming. IFN-γ, interferon-γ; JAK, Janus kinase; STAT, signal transducer and activator of transcription; P, phosphate group; PI3K, phosphoinositide 3-kinase; Akt, Ak strain transforming; mTOR, mammalian target of rapamycin; Foxp3, forkhead box P3; ROR-γt, retinoic acid-related orphan receptor gamma t; IDO, indoleamine 2,3-dioxygenase; TSG-6, tumour necrosis factor (TNF)-stimulated gene-6; TGF-β, transforming growth factor-beta; Smad, suppressor of mothers against decapentaplegic; HIF-1α, hypoxia-inducing factor-alpha; VEGF, vascular endothelial growth factor. Created with BioRender.com (accessed on 17 May 2024).

**Figure 3 biomedicines-12-01369-f003:**
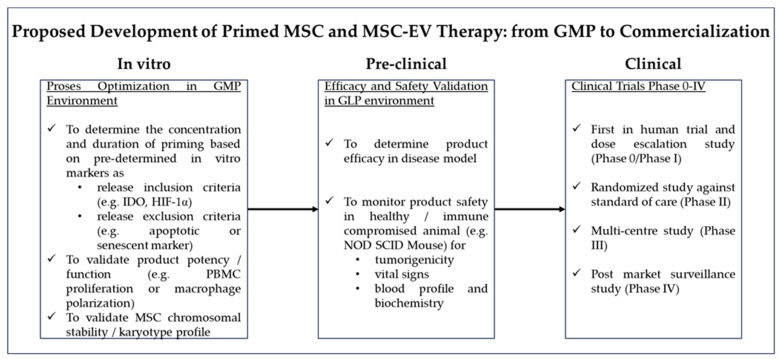
Flow chart for the systematic approach to developing primed MSC and EV therapies from GMP to commercialization. A structured approach has been proposed for optimizing MSC priming conditions by systematically adjusting both the concentration and duration of the priming agents to enhance the therapeutic efficacy of MSCs and their EVs. Primed MSCs and their EVs must fulfill inclusion and exclusion release criteria before proceeding to efficacy and safety validation in pre-clinical studies. Following this, a first-in-human trial for safety and dosage escalation is conducted before entering Phase II, Phase III and Phase IV clinical trials. GMP, Good manufacturing practice; GLP, Good laboratory practice; NOD/SCID, Nonobese diabetic/severe combined immunodeficiency.

**Table 1 biomedicines-12-01369-t001:** Pre-clinical studies reported priming with IFN-γ as a strategy for enhancing the therapeutic efficacy of MSC therapies.

Priming Concentration and Duration	Cell Sources	Pathological Condition and Recipients	Dosage	Route of Administration	In Vitro Markers	Functional Tests	Therapeutic Effects with Priming	Mechanisms of Action	Remarks	References
1 or 20 ng/mL recombinant mouse IFN-γ, 48 h	Mouse BM-MSCs	Systemic sclerosis (SSc) mouse model; immunocompetent mice	250 ng EVs; large-size EVs (lsEVs) or small-size EVs (ssEVs)	Intravenous injection	Cytokine level-HGF, IL1RA, PGE_2_ (ELISA), miRNA profile analysis (NGS)	Not stated	Improve remodelling and inflammatory mediators, reduce the expression of the fibrotic and inflammatory markers	Not stated (the upregulation of anti-inflammatory factors in MSCs via IFN-γ pre-activation was not observed in EVs, suggesting that other factors may be responsible for the improvement	Low-dose-primed EVs (derived from primed MSCs) were less efficient than naïve EVs (derived from naïve MSCs), while high-dose-primed EVs have similar efficacy to naïve EVs. Large-size EVs (lsEVs) were more efficient than small-size EVs (ssEVs), particularly in terms of improving remodelling and inflammatory markers in the skin.	Rozier et al., 2021 [35]
5 ng/mL IFN-γ + 5 ng/mL TNF-α, 12 h	Equine BM-MSCs	Osteoarthritis equine model; immunocompetent ponies	1 × 10^7^ MSCs	Intra-arterial injection	Not stated	Not stated	Reduce clinical and synovial inflammatory signs, as well as synovial effusion, improve cartilage macroscopic appearance and synovium histopathology	Via increasing anti-inflammatory effect	Despite the lack of significant differences between naïve MSCs and primed MSCs, the overall outcome suggested that primed MSCs had enhanced anti-inflammatory capabilities, with most improvements noted during the earlier time points.	Barrachina et al., 2018 [36]
10 ng/mL IFN-γ, 24 h	Human WJ-MSCs	Atopic dermatitis (AD) mouse model; immunocompetent mice	2 × 10^6^ MSCs	Subcutaneous injection	Gene expression profile (microarray analysis)	Not stated	Decrease immune cell infiltration, improve the features of AD, reduce epidermal and dermal thickness	Via the regulation of neutrophil-related Th17 immune responses	MSCs primed with IFN-γ elicited improved therapeutic effects in AD mice (better than non-primed MSCs).	Park et al., 2019 [37]
10 ng/mL IFN-γ + 10 ng/mL TNF-α, 24 h	Human UC-MSCs	LPS-induced neuroinflammation mouse model; immunocompetent mice	200 µL concentrated human UC-MSC-conditioned medium (injected twice on days 4 and 6)	Intravenous injection	Gene expression profile-*IDO1*, *CXCL9*, *IL6* (qRT-PCR), transcriptomic analysis (two-dimensional PCA, GO enrichment analysis)	Mouse microglial cell line -BV2 cell inhibition assay	Enhance the anti-inflammatory capacity, minimize the impact of donor-specific variations in MSC immunomodulation	Via stimulating IFN-γ and NF-κB signalling pathways	Priming MSCs improves immune suppressive function and reduces donor-dependent variations in immunomodulation. However, it hinders MSC proliferation. Thus, avoid priming during the expansion phase of MSCs.	Zhang et al., 2021 [22]
10 ng/mL IFN-γ + 10 ng/mL TNF-α, 24 h	Human UC-MSCs	Inflammatory bowel disease (IBD) mouse model; immunocompetent mice	1 × 10^6^ MSCs	Intravenous injection	Gene expression profile-*β-actin*, *CXCL9*, *CXCL10*, *CXCL11*, *IDO1*, *TSG-6* (qRT-PCR), protein level-IDO1, HA, COX2, AKT, HKII, β-actin, GLUT1(Western blot), TSG-6 (ELISA)	ECAR (glycolysis stress test kit), glucose uptake assay (flow cytometry), hexokinase activity assay (Hexokinase activity kit)	In vitro, there is an increase in glucose consumption and a metabolic shift towards glycolysis, while in vivo, there is a reduction in inflammatory parameters.	Via enhanced IDO and TSG-6 expression by promoting glycolysis, increasing glucose uptake and HKII activity	There is no naïve MSC group so no comparison was made. TNF-α and IFN-γ rapidly activate PI3K-AKT signalling promotes glycolysis in human MSCs, which is responsible for MSC-based anti-inflammatory therapy.	Xu et al., 2022 [38]
10 ng/mL IFN-γ, 48 h	Human BM-MSCs	Experimental autoimmune encephalomyelitis (EAE) mouse model; immunocompetent mice	1 × 10^6^ MSCs or 150 μg EVs derived from 5 to 7 × 10^6^ MSCs	Intravenous injection	MHC II, PD-L1 expression (flow cytometry), cytokine secretion-IDO, IL-6, etc. (luminex assay, ELISA)	T-cell (PBMC) proliferation (flow cytometry), Treg induction assay (flow cytometry)	Reduce demyelination and neuroinflammation	Via suppressing pathological T cell subset activation, inducing Tregs, probably via the “hit and run” mechanism	Exosomes were found to have similar efficacy to their MSC counterparts. IFN-γ priming ameliorated the disease to a higher extent than native MSCs and exosomes.	Riazifar et al., 2019 [39]
20 ng/mL IFN-γ + 50 μM KYNA, 24 h	Human AT-MSCs	2,4,6-trinitrobenzen sulfonic acid (TNBS)-induced acute colitis and chronic colon fibrosis rat model; immunocompetent rats	1.5 × 10^6^ MSCs (injected twice on days 1 and 3) for acute colitis; 2 × 10^6^ MSCs (injected three times for every two weeks) for chronic colon fibrosis	Intravenous injection	Transcriptome sequencing (GO enrichment analysis, KEGG enrichment analysis), IDO-1, iNOS, COX2 (qPCR, Western blot)	Not stated	Mitigate TNBS-induced colitis and colonic fibrosis, inhibit extracellular matrix (ECM) deposition and the EMT process, diminish the infiltration of inflammatory cells, suppress the inflammatory response, promote the polarization of M2 macrophages, and enhance homing to colon tissue	Via IFN-γ and KYNA-induced IDO-1, which facilitates cell homing, induces the polarization of intestinal macrophages to the anti-inflammatory M2 and elevates the expression of IL-10 to inhibit inflammation	MSCs primed with both IFN-γ and KYNA exhibit significant therapeutic efficacy in addressing acute colitis and chronic colon fibrosis in rats.	Ye et al., 2022 [40]
20 ng/mL IFN-γ, 48 h	Human UC-MSCs	EAE mouse model; immunocompetent mice	1 × 10^6^ MSCs	Intravenous injection	IDO1 expression (RT-PCR, Western blot)	Not stated	Mitigate body weight loss and clinical symptoms, reduce inflammation and latency of motor evoked potentials (MEP)	Via suppression of IL-17A and TNF-α expression, upregulation of IDO1	IFN-γ-MSCs showed more potent treatment efficacy than naïve MSCs.	Zhou et al., 2020 [41]
20 ng/mL IFN-γ, 48 h	Human UC-MSCs	EAE mouse model; immunocompetent mice	1 × 10^6^ MSCs	Intravenous injection	Not stated	Not stated	Alleviate body weight loss and clinical scores, regulate inflammation response	Via regulating the Th17/Tregs balance via the regulation of inflammatory cytokines production (increased IL-10 and decreased IL-17)	IFN-γ-MSCs showed more potent treatment efficacy than naïve MSCs.	Ling et al., 2022 [42]
25 ng/mL IFN-γ, 48 h	Human BM-MSCs	Sepsis mouse model; immunocompetent mice	1 × 10^6^ MSCs	Intravenous injection	Not stated	Not stated	Enhance microvascular hemodynamics during the initial stages of sepsis	Via increasing the red blood cell velocity, the rolling white blood cell flux and the number of venules with circulating white blood cells, the rate of soluble E-selectin (which may serve as a biomarker for monitoring endothelial damage in organs) was reduced	MSCs-IFN-γ appear to have a better beneficial effect on microvascular hemodynamics compared with naïve MSCs. However, the 6 h post-sepsis time point is too early to observe organ failures with this model.	Baudry et al., 2019 [43]
40 ng/mL IFN-γ, 24 h	Human UC-MSCs	Acute pneumonia mouse model; immunocompetent mice	1 × 10^6^ MSCs	Intravenous injection	Transcriptome sequencing (Illumina HiSeq 2500 system), IDO, PD-L1, JAK2, STAT1-3, etc. (Western blot)	T-cell (PBMC) proliferation (CFSE-flow cytometry), metabolism analysis (real-time assays of ECAR and OCR), metabolite analysis (LC-MS)	Preserve the alveolar structure in lung tissues, decrease the infiltration of inflammatory cells in alveoli, and lower levels of IL-1β and TNF-α	By redirecting the energy metabolism of MSCs towards aerobic oxidation and thus activates JAK-STAT signalling as well as induces IDO and PD-L1 production	Compared to the sole administration of IFN-γ, the combined treatment involving ATP and IFN-γ demonstrated better therapeutic efficacy. ATP amplifies the immunosuppressive capabilities of IFN-γ–primed MSCs by triggering the JAK-STAT pathway.	Yao et al., 2022 [8]
50 ng/mL IFN-γ, 8 h	Human UC-MSCs	*Escherichia coli*-induced pneumonia rat model; immunocompetent rats	1 × 10^8^ EVs, derived from 3.5 to 4 × 10^7^ MSCs	Intravenous injection	Not stated	Bacterial phagocytosis and killing assays (immunofluorescent)	Enhance survival rate, alleviate the severity of lung injury, modulate inflammatory response, reduce structural damage, restore lung structure	Via enhancement of macrophage phagocytosis and bacteria killing as well as restoration of endothelial nitric oxide synthase	IFN-γ-primed MSC-EVs were more effective than naïve MSC-EVs in reducing *E. coli*-induced lung injury.	Varkouhi et al., 2019 [44]
50 ng/mL IFN-γ, 24 h	Human BM-MSCs	Calvarial defect mouse model; both immunocompetent and immunodeficient mice	1–1.5 mm diameter MSC extracellular matrix (ECM) complex (C-MSCs)	Graft without artificial scaffold	IDO expression (RT-PCR, immunoblotting), IDO activity (level of kynurenine as a product of IDO-catabolism), osteogenic markers expression-*OPN*, *ALPase*, *BMP-2*, *OC* (RT-PCR)	T-cell (PBMC) proliferation (ELISA)	Induce bone regeneration, attenuate xenoreactive T cell response	By reducing the activity of xenoreactive T cells in mice via increased IDO expression and T cell suppression capacity in vitro, mainly via indirect paracrine effects rather than direct osteogenic differentiation	The use of primed C-MSCs was effective in preventing an unwanted immune response and promoting successful bone regeneration in immunocompetent mice, whereas the use of C-MSCs alone was not able to induce bone regeneration.	Takeshita et al., 2017 [45]
50 ng/mL IFN-γ + 50 ng/mL TNF-α, 24 h	Human BM-MSCs (aged human donors with end-stage OA)	Osteoarthritis (OA) mouse model; immunocompetent mice	2 × 10^4^ MSCs (without stimulation) or secretome from 2 × 10^4^ MSCs (with stimulation) (injected three times every two days)	Intra-articular injection	IDO activity (level of kynurenine as a product of IDO-catabolism). However, the comparison was not clear	Not stated	Early pain reduction, protect against cartilage damage	Not stated (no significant results observed to support any conclusions)	Secretome from stimulated MSCs diminished pain and OA-related structural changes, and these effects were at least as effective as the injection of MSCs without stimulation.	Khatab et al., 2018 [46]
50 ng/mL IFN-γ, not stated	Mouse BM-MSCs	Dextran sulphate sodium (DSS)- induced colitis mouse model; immunocompetent mice	200 μg EVs	Intravenous injection	EVs number (EXOCET exosome quantitation kit), EVs protein-CD9, CD81 (Western blot)	Not stated	Restore body weight loss and impaired intestinal structure, decrease disease activity index (DAI) score and colon shortening, reduce inflammation cytokines	Via inhibiting Th17 cell differentiation via increased levels of *miR-125a* and *miR-125b* in EVs, promoting Treg cell differentiation	Priming MSCs with IFN-γ generated EVs with better anti-colitis therapeutic efficacy.	Yang et al., 2020 [47]
100 ng/mL IFN-γ, 48 h	Human AT-MSCs	Carbon tetrachloride (CCl_4_)-induced liver cirrhosis mouse model; immunocompetent mice	2 μg or 5 μg EVs	Intravenous injection	Proteomics (nano-LC-MS) and miRNA content analysis (DNAFORM) of EVs	Macrophage polarity-mRNA expression of genes encoding pro- and anti-inflammatory factors (qPCR), motility and phagocytosis (immunofluorescent) assays	Ameliorate fibrosis and inflammation, promote tissue repair	Via the induction of anti-inflammatory macrophages with higher motility and phagocytic ability, increasing regulatory T cell counts	IFN-γ priming resulted in enhanced efficacy of MSC-derived EVs.	Takeuchi et al., 2021 [48]
200 IU/mL IFN-γ, 24 h	Human BM-MSCs, CB-MSCs, AT-MSCs, WJ-MSCs	Graft-versus-host disease (GVHD) mouse model; immunodeficient mice	1 × 10^6^ MSCs (injected twice on days 0 and 7)	Intravenous injection	Gene expression profile-*CXCL9*, *CXCL10*, *CCL8*, *IDO*, etc. (RT-PCR, qRT-PCR, microarray analysis)	T-cell (PBMC) proliferation (BrdU Incorporation Assay)	Improve survival rate, decrease clinical symptoms and immune cell infiltration	Via the suppression of antigen-driven proliferation of T-cells	IFN-γ-primed MSCs showed better therapeutic efficacy than naïve MSCs. The effect between different MSC sources was not compared.	Kim et al., 2018 [49]
500 U/mL IFN-γ + 5000 U/mL TNF-α, 24 h	Mouse BM-MSCs	Diabetic mouse model; immunocompetent mice	500 IEQ of encapsulated islets with MSCs at a 1:1 ratio	Intraperitoneal injection	Cytokine/chemokine secretion-CXCL9, CXCL10, IL-6, COX-2, IDO, etc. (RT-PCR, cytokine protein array panel), nitric oxide (NO) production (RT-PCR, nitrite/nitrate (NO_2_/NO_3_) colorimetric kit)	Not stated	Normalize blood glucose levels, reduce pericapsular fibrotic overgrowth (PFO)	By increasing the expression of anti-inflammatory cytokines such as *IL-4*, *IL-6*, *IL-10*, and *G-CSF*, as well as boosting NO production, which are known to regulate the immune response	IFN-γ, in combination with TNF-α, synergistically enhanced the immunosuppressive effects of murine MSCs. Neither IFN-γ nor TNF-α are sufficient on their own.	Vaithilingam et al., 2017 [50]
500 U/mL recombinant human IFN-γ, 48 h	Human BM-MSCs	Humanized GVHD mouse model; Immunodeficient mice	4.4 × 10^4^ MSCs per gram	Intravenous injection	Not related to the in vitro functional tests that used naïve MSCs instead of primed MSCs	Not related	Extend lifespan, decrease liver and gut pathology but not lung pathology	By directly inhibiting the proliferation of donor T cells and decreasing the production of human TNF-α from T cells	Therapeutic efficacy of IFN-γ stimulated MSCs on day 0 is comparable to that of unstimulated MSCs on day 7, where MSCs require IFN-γ pre-stimulation for efficacy at the earliest time points.	Tobin et al., 2013 [51]
500 U/mL recombinant human IFN-γ, 72 h	Human WJ-MSCs	EAE mouse model; immunocompetent mice	1 × 10^6^ MSCs (injected twice on days 3 and 11 post-immunization)	Intravenous injection	Cytokine level-VEGF, IL-10, TGF-β, HGF (ELISA)	PBMC proliferation (CFSE-flow cytometry)	Reduce T cell reactivity, enhance neurological functional recovery, reduce infiltration of inflammatory cells, delay the onset of clinical symptom	Via modulating immune differentiation from a Th1 towards a Th2 phenotype, increasing the frequency of CD4^+^CD25^+^CD127^low/neg^ Foxp3^+^T regulatory cells	IFN-γ-primed MSCs have better immunomodulatory function than unprimed MSCs.	Torkaman et al., 2017 [52]
500 U/mL IFN-γ, 7 days	Human BM-MSCs	Myocardial infarction (MI) mouse model; Immunodeficient mice	2 × 10^5^ MSCs	Intramyocardial injection	IDO expression (immunohistochemistry). However, no comparison was made between naïve MSC and primed MSC	PBMC proliferation (3H-thymidine uptake in counts per minute (CCPM)). However, no comparison was made	Neither unstimulated MSC therapy nor IFN-γ-stimulated MSC therapy shows any significant positive impact on cardiac function or remodelling	Not stated	Both MSCs engraft in infarct myocardium. The animal models used for cardiac MSC therapy seem to be less strong than originally anticipated.	Haan et al., 2016 [53]

BM—Bone marrow; MSC—Mesenchymal stem cell; IFN-γ—Interferon-gamma; TNF-α—Tumour necrosis factor-alpha; IDO—Indoleamine 2,3-dioxygenase; PBMC—Peripheral blood mononuclear cell; RT-PCR—Reverse transcription-polymerase chain reaction; *OPN*—*Osteopontin; ALPase*—*Alkaline phosphatase*; *BMP-2*—*Bone morphogenetic protein-2*; *OC*—*Osteocalcin*; ELISA—Enzyme-linked immunosorbent assay; WJ—Wharton’s jelly; VEGF—Vascular endothelial growth factor; IL—Interleukin; TGF-β—Transforming growth factor-beta; HGF—Hepatocyte growth factor; CFSE—Carboxyfluorescein diacetate succinimidyl ester; Th1—T helper type 1; Th2—T helper type 2; CD—Clusters of differentiation; Foxp3—Forkhead box P3; IEQ—Islet equivalent; CXCL—Chemokine (C-X-C motif) ligand; COX-2—Cyclooxygenase-2; G-CSF—Granulocyte colony-stimulating factor; CB—Umbilical cord blood; AT—Adipose tissues; *CCL8*—*Chemokine* (*C-C motif*) *ligand 8*; qRT-PCR—Real-time quantitative RT-PCR; BrdU—Bromodeoxyuridine; EVs—Extracellular vesicles; MHC II—Major histocompatibility complex class II; PD-L1—Programmed death-ligand 1; Treg—Regulatory T cells; UC—Umbilical cord; miR-125—MicroRNA-125; IL1RA—Interleukin-1 receptor antagonist; PGE_2_—Prostaglandin E2; NGS—Next-generation sequencing; nano-LC-MS—Nanoscale liquid chromatography-coupled mass spectrometry; qPCR—Quantitative PCR; LPS—Lipopolysaccharide; PCA—Principle Component Analysis; GO—Gene ontology; *TSG-6*—*TNF stimulated gene-6*; HA—Hemagglutinin; AKT—Protein kinase B; HKII—Hexokinase II; GLUT1—Glucose transporter 1; ECAR—Extracellular acidification; JAK—Janus kinase; STAT—Signal transducer and activator of transcription; OCR—Oxygen consumption rate; ATP—Adenosine triphosphate; KYNA—Kynurenic acid; KEGG—Kyoto Encyclopaedia of Genes and Genomes; iNOS—Inducible nitric oxide synthase; EMT—Epithelial–mesenchymal transition; NF-κB—Nuclear factor kappa light chain enhancer of activated B cells; PI3K—Phosphoinositide 3-kinase.

**Table 2 biomedicines-12-01369-t002:** Pre-clinical studies reported priming with hypoxia as a strategy for enhancing the therapeutic efficacy of MSC therapies.

Priming Method and Duration	Cell Sources	Pathological Condition and Recipients	Dosage	Route of Administration	In Vitro Markers	Functional Tests	Therapeutic Effects of Priming	Mechanisms of Action	Remarks	References
Hypoxic condition (0.5% O_2_), 24 h	Mouse BM-MSCs	MI mouse model; immunocompetent mice	2 × 10^5^ MSCs	Intramyocardial injection	hLeptin expression (qRT-PCR, ELISA), mCXCR4 (qRT-PCR, flow cytometry)	MSC migration assay (Trans-well system), MSC apoptosis assay (TUNEL staining), tube formation assay (HUVEC model), cardio-protection assay (TUNEL staining)	Enhance cell homing and survival rate, improve systolic function and cardiac function, promote angiogenesis, reduce apoptosis rate	Via enhanced leptin expression	Hypoxia-induced expression of leptin plays a crucial role in the protective effects of hypoxic MSCs. Hypoxic MSCs showed significantly better therapeutic efficacy than normoxic MSCs.	Hu et al., 2014 [85]
Hypoxic condition (1% O_2_), 24 h	Human AT-MSCs	Acute kidney injury (AKI/renal IRI) rat model; immunocompetent rats	2 × 10^6^ MSCs	Injected into the left kidney cortex	Growth factor assay-bFGF, VEGF, HGF (RT-PCR, ELISA)	Cell viability assay (flow cytometry-propidium iodide (PI) solution)	Enhance antioxidative capacity and angiogenic effects, decrease apoptosis rate, attenuate renal injury, improve renal function	Via improved activated paracrine effects (secretion of angiogenic factors)	Hypoxia preconditioning significantly enhanced the therapeutic effects of MSCs on AKI.	Zhang et al., 2014 [78]
Hypoxic condition (1% O_2_), 24 h	Rat or Human BM-MSCs	Renal ischemia-reperfusion injury (IRI) rat model; immunocompetent rats	5 × 10^5^ MSCs	Injected through the abdominal aorta clamped above and below the left renal artery bifurcation	Phosphorylated Smad2 (pSmad2) and α-SMA expression (Western blot), VEGF, HGF, PGE_2_ levels (ELISA), *VEGF*, *HGF* expression (qRT-PCR)	Not stated	Attenuate IRI-induced renal fibrosis, suppress the infiltration of inflammatory cells	Via inhibition of TGF-β/Smad signalling and upregulation of VEGF expression (renal fibrosis attenuation), augmentation of PGE_2_ secretion (immunomodulation)	Hypoxic MSCs significantly ameliorate renal fibrosis and inflammation in IRI rats compared to normoxic MSCs. Hypoxic preconditioning does not increase the engraftment capacity of MSCs.	Ishiuchi et al., 2020 [76]
Hypoxic condition (1% O_2_), 24 h	Human BM-MSCs	Renal IRI rat model; immunocompetent rats	5 × 10^5^ MSCs	Injected into the abdominal aorta clamped above and below the left renal artery bifurcation	Phosphorylated Smad2 (pSmad2) and α-SMA expression (Western blot), VEGF and HGF secretion (ELISA)	Cell proliferation assay (WST-1 assay), migration assay (Trans-well system), macrophage polarization assay (Western blot and qRT-PCR)	Attenuate IRI-induced renal fibrosis, suppress inflammatory cell infiltration	Via the stimulation of HGF secretion and subsequently enhancing the inhibition of the TGF-β/Smad signalling pathway, the enhancement of paracrine activity without conflicting interactions	Serum-free medium and hypoxia preconditioning synergistically enhanced MSCs’ proliferative capacity and efficacy in attenuating renal fibrosis injury, whereas the anti-inflammatory effect of normoxic MSCs is almost equal to hypoxic MSCs.	Ishiuchi et al., 2021 [77]
Hypoxic condition (1% O_2_), 48 h	Porcine AT-MSCs	Atherosclerotic renal artery stenosis (ARAS) porcine model	1 × 10^7^ MSCs	Intra-arterial injection (injected into renal arteries)	Not stated	Not stated	Reduce diastolic blood pressure, restore renal medullary oxygenation, decrease kidney fibrosis and interstitial T-cell infiltration	Not stated	Hypoxia preconditioning of AT-MSCs was comparable to normoxia and did not improve the effects of MSC on renal function despite inducing a reduction in DNA hydroxymethylation (5hmC levels) of inflammatory and profibrotic genes.	Farooqui et al., 2023 [31]
Hypoxic condition (1% O_2_), 48 h	Human MSC cell line (HUM-iCELL-e009)	Transverse aortic constriction (TAC) mouse model; immunocompetent mice	200 µg EVs (injected twice on days 7 and 14)	Intravenous injection	PARK7/DJ-1 protein levels (quantitative proteomics analysis)	OCR (Seahorse Extracellular Flux Analyzer), mitochondrial membrane potential (JC-1 kits), ROS levels (MitoSOX™ Red reagent), ubiquitylation assays (Western blot), mtDNA copy number (qPCR)	Alleviate myocardial hypertrophy, demonstrate cardioprotective properties, improve mitochondrial function	Via the expression of DJ-1 in hypoxic ps leads to the alleviation of mitochondrial damage and suppression of ATRAP degradation.	Hypoxic EVs demonstrate a heightened inhibitory impact on cardiac hypertrophy in comparison to normoxic EVs.	Lu et al., 2023 [86]
Hypoxic condition (1% O_2_) throughout the whole culturation process	Human UC-MSCs	OVA-induced allergic rhinitis (AR) mouse model; immunocompetent mice	100 µg CM or EVs (injected six times for every three days)	Intravenous injection	*OCT4*, *SOX2*, *Nanog* expression (qRT-PCR), Telomerase activity (PCR-ELISA), VEGF level (Western blot)	Evaluation of maturation status of dendritic cells (flow cytometry)	Reduce the frequency of sneezing and scratching, suppress allergic inflammation, and support remodelling of the nasal mucosa	By increasing VEGF levels in hypoxic EVs that suppressed the differentiation and maturation of dendritic cells	Prolonged exposure to hypoxia improves the proliferative capacity, enhances telomerase activities, reduces senescence, and preserves the multipotent status of UC-MSCs compared to normoxic conditions. Notably, both hypoxic EVs and CM exhibit better therapeutic effects compared to their normoxic counterparts, implying that the essential components in the CM likely originate from EVs.	Wu et al., 2023 [30]
Hypoxic condition (3% O_2_) + 1.8 mM calcium throughout the whole culturation process	Human UCB-MSCs	Humanized graft-versus-host disease (GVHD) mouse model; immunodeficient mice	5 × 10^5^ SHC-MSCs (Small MSCs primed with hypoxia and calcium ions)	Intravenous injection	Genome-wide gene expression and DNA methylation analyses (microarray analysis/gene set enrichment analysis)	Anti-inflammation assay secretion of TNF-α (ELISA), MLR assay (Human T-cell (PBMC) proliferation), tube formation assay (HUVEC model)	Increase survival rate, attenuate weight loss, decrease immune cell infiltration and characteristic tissue injuries, enhance tissue repair and cell homing, reduce inflammatory cytokines level	Via the overexpression of *PLK1*, *ZNF143*, *FOG*, and *DHRS3* and subsequently enhance the proliferative, self-renewal, migratory, pro-angiogenic, anti-inflammatory, and immunomodulatory capacities of MSCs	SHC-MSCs have an enhanced potency for treating GVHD compared to the naïve MSCs.	Kim et al., 2018 [33]
Hypoxic condition (5% O_2_), 24 h	Human AT-MSCs	Traumatic brain injury (TBI) rat model; immunocompetent rats	0.1 mL/250 g secretome (ST) (once daily for 7 days)	Intravenous injection	Proteome analysis (LC-MS/MS), VEGF, BDNF, GDNF, PDGF-BB, ICAM-1 concentration (ELISA)	Not stated	Mitigate neurological impairment and cognitive deficiency, alleviate neuroinflammatory edema, reduce nerve fibre damage, improve neuroinflammatory environment, limit the apoptosis of neural cells	Via the mediation of secondary neuroinflammation, it fosters the polarization of microglia into M2 phenotypes (anti-inflammatory)	There is no normoxic MSC group while serum-free basal media was injected as comparison. Hypoxic MSC-ST can improve neural functional outcomes after TBI in its early stages by mediating secondary neuroinflammation.	Xu et al., 2020 [80]
Hypoxic condition (5% O_2_), 24 h	Human UC-MSCs	OVA-induced chronic asthma mouse model; immunocompetent mice	40 µg EVs (inhaled or injected four times for every seven days)	Inhalation or intravenous injection	Not stated	Not stated	Reduce chronic airway inflammation, inhibit the prevailing type-2 immune response, and deter airway remodelling	Not stated	The non-invasive approach of nebulized hypoxic EVs inhalation significantly decreases chronic airway inflammation and remodelling, with the EVs predominantly accumulating in the lungs and maintaining their presence for a period of 7 days.	Xu et al., 2023 [32]
Hypoxic condition (5% O_2_), 1000 units of IFN-γ, 72 h	Rat BM-MSCs	Rat hindlimb allotransplantation model; immunocompetent rats	2 × 10^6^ MSCs	Ex vivo allograft engineering (vasculature was perfused with BM-MSCs in cold media, divided between 3 separate perfusates over the course of 1 h)	*IDO* expression (qRT-PCR)	CD8^+^ T cell-mediated cytotoxicity assay (lactate dehydrogenase-based), CD4^+^ T cell proliferation assay (3H-thymidine-based), migration assay (Trans-well system), cell proliferation assay (MTT Cell Proliferation Assay Kit)	Postpones the onset of acute rejection while preserving the recipient’s adaptive immune response	By elevating the expression of *IDO*, it provides a shield for endothelial cells, hinders the proliferation of CD4^+^ T cells, and enhances cell motility and proliferative potential	Hypoxic priming is significantly better than IFN-γ priming in prolonging allograft rejection. Both primed MSCs were better than unprimed MSCs.	Soares et al., 2018 [79]
Hypoxic condition (5% O_2_), 72 h	Human UC-MSCs	Cutaneous-wound healing in a diabetic rat model; immunocompetent rats	0.5 mL of undiluted CM	Intradermal injection on a peripheral wound	VEGF, bFGF, pro-collagen 1 secretion (ELISA)	Rat fibroblast cell growth (CCK-8 viability assay), collagen production analysis on rat fibroblasts (ELISA)	Facilitate wound closure and re-epithelialization	Not stated	Hypoxic MSC-CM treatment showed a distinct effect in facilitating wound repair in the early stage of the diabetic wound model compared to the topical antibiotic treatment (bactoderm mupirocin ointment 2%).	Hendrawan et al., 2021 [87]

BM—Bone marrow; MSC—Mesenchymal stem cell; O_2_—Oxygen; MI—Myocardial infarction; qRT-PCR—Real-time quantitative reverse transcription-polymerase chain reaction; ELISA—Enzyme-linked immunosorbent assay; CXCR4—C-X-C chemokine receptor type 4; TUNEL—Terminal deoxynucleotidyl transferase-mediated dUTP nick end-labelling; HUVECs—Human umbilical vein endothelial cells; AT—Adipose tissues; IRI—Ischemia-reperfusion injury; bFGF—Basic fibroblast growth factor; VEGF—Vascular endothelial growth factor; HGF—Hepatocyte growth factor; UCB—Umbilical cord blood; DNA—Deoxyribonucleic acid; TNF-α—Tumour necrosis factor-alpha; MLR—Mixed lymphocyte reaction; PBMC—Peripheral blood mononuclear cell; *PLK1*—Polo-like kinase-1; *ZNF143*—Zinc-finger protein-143; *FOG*—Friend-of-GATA; *DHRS3*—Dehydrogenase/reductase-3; IFN-γ—Interferon-gamma; IDO—Indoleamine 2,3-dioxygenase; CD—Clusters of differentiation; MTT—3-[4,5-dimethylthiazol-2-yl]-2,5 diphenyl tetrazolium bromide; Smad—Suppressor of mothers against decapentaplegic; α-SMA—Alpha-smooth muscle actin; PGE_2_—Prostaglandin E2; TGF-β—Transforming growth factor-beta; LC-MS/MS—Liquid chromatography–tandem mass spectrometry; BDNF—Brain-derived neurotrophic factor; GDNF—Glial cell line-derived neurotrophic factor; PDGF-BB—Platelet-derived growth factor; ICAM-1—Intercellular cell adhesion molecule-1; UC—Umbilical cord; CCK-8—Cell Counting Kit-8; WST-1—Water-soluble tetrazolium salt-1; OVA—Ovalbumin; EVs—Extracellular vesicles; CM—Conditioned medium; *OCT4*—Octamer-binding transcription factor 4; *SOX2*—Sex determining region Y-box 2; PARK7/DJ-1—Parkinson disease protein 7; OCR—Oxygen consumption rate; JC-1—tetraethylbenzimidazolylcarbocyanine iodide; ROS—Reactive oxygen species; mtDNA—Mitochondrial DNA; ATRAP—AT1R-associated protein.

## Data Availability

No new data were created or analyzed in this study. Data sharing is not applicable to this article.

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
