# Peer review of "Therapeutic Efficacy of Interferon-Gamma and Hypoxia-Primed Mesenchymal Stromal Cells and Their Extracellular Vesicles: Underlying Mechanisms and Potentials in Clinical Translation"

_biomedicines, 2024, doi:10.3390/biomedicines12061369_

Round 1
Reviewer 1 Report
Comments and Suggestions for Authors
Multipotent mesenchymal stem/stromal cells (MSCs) from various tissue sources are currently considered as a promising tool for regenerative medicine and cell therapy.
MSCs have unique properties such as increased proliferation,multilineage differentiation and paracrine activity including production of immunomodulatory mediators. The elucidation of the molecular mechanisms of MSCs’ action as well as enhancement of MSCs’ clinically-relative potential are on demand in current research. The priming of MSCs by hypoxia and inflammatory mediators and the mechanisms involved in these effects are on request for the MSC preconditioning protocols for application in regenerative medicine. Thus, the updating of state-of-art in this field could be very interesting and useful for people dealing with these area of MSC biology.
“Mesenchymal Stromal Cells” in the Title and further. The term “mesenchymal stromal cells” covers all cells of stromal lineage including not only stem/progenitor cells, but terminally differentiated as well (like osteocytes, chondrocytes, etc.) It will be more correct to name these cells as multipotent mesenchymal stem/stromal cells (MSCs) when mentioned for the first time.
The authors have reviewed the papers describing inflammatory priming to enhance the synthesis of immunomodulatory factors in MSCs, and I hypoxic priming - to stimulate growth factors and cytokines that promote regeneration. However, there are studies that show that inflammatory cytokines can induce the production of growth factors in MSCs. Besides, the increase of the immunosuppressive potential of MSCs under hypoxia has been demonstrated as well. Perhaps, the authors should make some “crossroads” in the description of priming functions of these two factors. Additionally, it would be advantageous to expand the summary table on priming effects of IFN-gamma. For instance, the review " Inflammatory priming of mesenchymal stem cells: Focus on growth factors enhancement " provides a comprehensive overview of the subject matter. The papers that focus on the enhancement of growth factors describe the secretion of various growth factors, including VEGF, G-CSF, BDNF, NGF, and IGF-1, by MSCs in response to the use of proinflammatory cytokines such as IFN-γ, IL-1β, and TNF-α.
LL49-50 “MSCs derived from different tissue sources displayed varying properties in terms of cellular composition…” What did the authors mean under cellular composition of MSCs?
LL57-64. This fragment is somewhat disorganized and requires careful rewriting. Concerning three aspects of MSC-based therapy: Why have the authors considered the “cell replacement involves MSCs differentiating into various cell types to replace damaged tissues” as a first point. Please provide relevant references to exemplify this point. At a moment, this path is not considered the first one. Further, the authors have mentioned immunomodulation as a second path. It is unclear why this aspect was considered separately from MSC secretory activity? A similar question can be posed regarding the EVs story, which also result from MSC secretion.
L246 Please, clarify how are VEGF and HGF involved in immunosuppression?
The authors correctly identified IFN-γ and TNF-α as the two primary proinflammatory cytokines that stimulate the immunomodulatory activity of MSCs. However, it is unclear why IFN-γ was selected for the review in particular.
Table 1. The authors' mention of pre-activated EVs suggests that EVs are activated separately from MSCs. It should be noted that this is a product of primed MSCs.
The same is related to “6. Interferon-γ (IFN- γ) Priming of MSC Derived Extracellular Vesicles (MSC-EVs)” and Figure 1
Reviewer 2 Report
Comments and Suggestions for Authors
The manuscript "Therapeutic Efficacy of Interferon-Gamma and Hypoxia Primed Mesenchymal Stromal Cells and Their Extracellular Vesicles: Underlying Mechanisms and Potentials in Clinical Translation" reviews the enhanced therapeutic potentials of Mesenchymal Stromal Cells (MSCs) and their Extracellular Vesicles (EVs) when primed with interferon-gamma (IFN-γ) and hypoxia. It focuses on how these priming conditions affect MSCs' immunomodulatory, angiogenic, proliferation, and tissue regeneration capabilities, driven primarily by JAK/STAT and PI3K/AKT signaling pathways for IFN-γ, and Leptin/JAK/STAT and TGF-β/Smad for hypoxia . Here is a detailed review of the manuscript:
1. Clarification of MSC Source Variation: The manuscript discusses results from various sources of MSCs, including bone marrow, adipose tissue, and umbilical cord. It is crucial to clarify how the source of MSCs might influence the results and whether the findings are broadly applicable across all sources or specific to one type.
2. Standardization of Priming Protocols: The manuscript points out the lack of standardized protocols in MSC priming. It would be beneficial to propose a more structured approach to developing these standards, perhaps suggesting specific methodologies that have shown promising repeatability in results .
3. Comparative Analysis of Priming Methods: While both IFN-γ and hypoxia are discussed, a comparative analysis between these methods and possibly others in terms of efficacy and safety could enrich the discussion. This might include a table or a systematic review of studies comparing different priming agents.
4. Long-term Effects and Safety: The manuscript should address the long-term therapeutic safety and potential adverse effects of primed MSCs and EVs. Including a section on post-treatment follow-ups in clinical trials could provide deeper insight into the longevity of the benefits and any delayed adverse effects.
5. Mechanistic Insights at the Cellular Level: Further elucidation on the signaling pathways and their interactions would enhance the understanding of how priming affects MSCs at the molecular level. This could include detailed diagrams or models based on current hypotheses or findings.
6. Scale-up and Manufacturing Challenges: There is mention of up-scaling production, but specific challenges associated with scaling up these priming techniques for clinical applications should be discussed more thoroughly. This could involve a discussion on the bioreactor designs, control of culture conditions, and regulatory considerations.
7. Ethical and Regulatory Considerations: While briefly mentioned, a detailed section on ethical issues, particularly concerning the source of MSCs and the regulatory pathways for clinical application of such primed cells and EVs, would be a crucial addition to ensure the manuscript covers all aspects necessary for translation to clinical practice.
The reviewer suggests the following articles related to the application of msc-evs for the authors to refer to in order to revise the manuscript:
1. BiomaterTransl. 2022Sep28;3(3):175-187.doi:10.12336/biomatertransl.2022.03.002. eCollection 2022.
Comments on the Quality of English Language
The writing quality of some English sentences in this paper can be further improved, and it is suggested that appropriate polishing should be done.
Reviewer 3 Report
Comments and Suggestions for Authors
The authors have conducted a review on the main MSCs priming protocols, with a focus on IFN-gamma and hypoxia priming. This review, although not a systematic one, has been conducted with rigour; however, there are a few things that need to be assessed, in order to improve the quality of this work.
First, the introduction section could benefit to be shortened a bit, as the importance of priming MSCs with IFN-gamma and hypoxia is explained in a detailed matter throughout the rest of the text, so that a briefer introduction of this matter should be preferred.
The authors have only briefly mentioned the potential of MSCs in the treatment of diabetes, which is a topic of great importance, given the vast population that is affected by this disease. Perhaps they could implement this aspect, including some recent works on the role of MSCs in general and of primed MSCs in the treatment of Type 1 diabetes mellitus (DOI: 10.1002/smll.202301019; doi: 10.1210/clinem/dgad142; doi: 10.7150/ijms.87472; DOI: 10.3390/biomedicines11051426; DOI: 10.3389/fimmu.2021.732549).
Figure 1: I would suggest that the authors justify margins in the written section, as right now the figure looks a bit messy. It would also be better to put more emphasis on the figure itself (by making it bigger, and/or by reducing the font size of the written parts).
In general, the authors should write the full name of growth factors, cytokines, and so on, only the first time they mention it, followed by the abbreviation, and then continue to use the abbreviation alone during the text in all the paragraphs, without specifying the full name again, in order to ease the reader.
Comments on the Quality of English LanguageThe authors should carefully check for proper English structure and grammar throughout the text (i.e., the sentence in line 123 should be revised, the sentence in line 700 – “While SDF-1/CXCR4 axis promotes cell homing and survival.” is a stand-alone subordinate that should probably be connected to the previous sentence).
Round 2
Reviewer 3 Report
Comments and Suggestions for Authors
The authors have addressed most of my suggestions, and the quality of their work has certainly improved. However, as previously mentioned, please include the following articles to the review, as they represent some of the most recent publications on the topic of MSCs and DM (they have been published in the last year, while the authors have only added older publications):
- DOI: 10.3390/biomedicines11051426
- DOI: 10.1002/smll.202301019
Author Response
Thank you for your suggestions. The publications have been discussed and included in the manuscript. Please refer to lines 295-312.
Other than that, numerous studies have harnessed the immunomodulatory property of MSCs for treating Type I diabetes, which is a well-known autoimmune disease characterized by specific adaptive immunity against β-cell antigens (46,63,64). For instance, Wang et al demonstrated that cytokine-primed MSC-EVs exhibited high levels of the immune checkpoint molecule PD-L1, which significantly reduced CD4+ T cell density and activation through the PD-L1/PD-1 axis and promoted the transition of macrophages from the M1 to M2 phenotype in mice with Type I diabetes (64). Pancreatic islet transplantation is a therapeutic option for treating Type I diabetes; however, acute islet loss is a significant complication of this procedure. Barachini et al reviewed various approaches using MSCs and MSC-EVs to create a more conducive immune microenvironment, aiming to reduce graft rejection and promote early vascularization to support graft survival (65). Mrahleh et al reported that MSCs primed with IFN-γ and TNF-α exhibited immunomodulatory effect on CD4+ and CD8+ T cells by producing tolerogenic dendritic cells which inhibit antigen-specific T cell responses through induction T cell anergy (63). Similarly, Vaithilingam et al reported improved encapsulated islet allograft survival and function through both co-encapsulation and co-transplantation of islets with primed MSC (46). These findings have great implication in the future management and treatment of diabetes that affects millions of patients worldwide (66).